# A secreted metal-binding protein protects necrotrophic phytopathogens from reactive oxygen species

Lulu Liu[1,5], Virginie Gueguen-Chaignon[2,5], Isabelle R Gonçalves[1,5], Christine Rascle[1], Martine Rigault[3], Alia Dellagi[3], Elise Loisel[1], Nathalie Poussereau[1], Agnès Rodrigue [1], Laurent Terradot [4]* & Guy Condemine [1]*

Few secreted proteins involved in plant infection common to necrotrophic bacteria, fungi and oomycetes have been identified except for plant cell wall-degrading enzymes. Here we study a family of iron-binding proteins that is present in Gram-negative and Gram-positive bacteria, fungi, oomycetes and some animals. Homolog proteins in the phytopathogenic bacterium *Dickeya dadantii* (IbpS) and the fungal necrotroph *Botrytis cinerea* (BcIbp) are involved in plant infection. IbpS is secreted, can bind iron and copper, and protects the bacteria against $H_2O_2$-induced death. Its 1.7 Å crystal structure reveals a classical Venus Fly trap fold that forms dimers in solution and in the crystal. We propose that secreted Ibp proteins binds exogenous metals and thus limit intracellular metal accumulation and ROS formation in the microorganisms.

[1] Microbiologie Adaptation et Pathogénie, UMR 5240 CNRS, Université de Lyon, INSA de Lyon, 69622 Villeurbanne, France. [2] Protein Science Facility, SFR BioSciences, UMS3444/US8, 69367 Lyon, France. [3] Institut Jean-Pierre Bourgin, UMR1318 INRA-AgroParisTech, 78026 Versailles, France. [4] Molecular Microbiology and Structural Biochemistry, UMR 5086 CNRS, Institut de Biologie et Chimie des Protéines, Université de Lyon, 69367 Lyon, France. [5]These authors contributed equally: Lulu Liu, Virginie Gueguen-Chaignon, Isabelle R Gonçalves. *email: laurent.terradot@ibcp.fr; guy.condemine@insa-lyon.fr

Plants are regularly exposed to biotic stresses causing important economic losses. Major diseases are caused by phytopathogenic bacteria, viruses, fungi or oomycetes owing to infectious strategies developed by the parasites that enable them to feed and grow on their hosts. To undergo their infectious cycle, plant pathogens adopt diverse parasitic lifestyles. For example, biotrophs develop and feed on living tissues by maintaining a tightly regulated interaction in which the microbe does not kill its host[1]. Necrotrophs feed on dead tissues after inducing their necrosis or collapse[2] and hemibiotrophs evolve as biotrophs during the early stages of infection and then shift to a necrotrophic phase[3]. The secretion of host-selective or host-nonselective toxins and of plant cell wall-degrading enzymes are among the common strategies deployed by necrotrophic and hemibiotrophic phytopathogens[4,5] that contribute to the depolymerization of the structural plant cell wall polysaccharide components. Among all characterized proteins secreted by phytopathogenic microbes, only Nep1-like proteins (NLP) have been detected in fungi, oomycetes and bacteria[6]. These proteins trigger leaf necrosis and plant immunity-associated defenses in dicotyledonous plants[7]. The widespread occurrence of this protein family in phytopathogenic microbes highlights their importance for the pathogens and a clear virulence role for this family has been observed in the bacterium *Pectobacterium carotovorum* and the fungus *Verticillium dalhiae*[8].

Upon infection, plants can activate a large array of defense responses, regardless of the infectious parasites[9]. Effective defenses involve the production of reactive oxygen species (ROS), cell wall strengthening, and the production of phytoalexins and pathogenesis-related (PR) proteins. Plants are equipped with a basal immune system called pattern triggered immunity (PTI) that is based on the recognition of conserved motifs, called microbe-associated molecular patterns followed by activation of the abovementioned defenses[10]. Some pathogens can secrete effector proteins to suppress PTI. Plants have evolved resistance proteins capable of detecting such effectors, thereby triggering strong defenses that culminate in a controlled, localized cell death called hypersensitive response. This strong immune response is termed effector triggered immunity (ETI).

ROS production comprises a general response occurring during either PTI or ETI. During infection, the superoxide anion produced by a plasma membrane-anchored NADPH oxidase is the first ROS source, followed by a rapid dismutation of the superoxide anion into hydrogen peroxide[11]. In the presence of the redox-active metals $Fe^{2+}$ and $Cu^{2+}$, the Fenton reaction produces the highly toxic hydroxyl radical, which can severely damage macromolecules such as proteins, DNA, and lipids. Although copper and iron can catalyze the formation of highly toxic ROS, these transition metals are essential for most living organisms as they function as cofactors of enzymes involved in key redox reactions. Because of this dual property, $Cu^{2+}$ and $Fe^{2+}$ homeostasis is tightly regulated in all organisms to ensure their physiological needs and avoid toxicity. During a host–pathogen interaction, the management of these metals by both partners is complicated[12–14] and the oxidative stress generated may have deleterious effects on both the host and the pathogen. Thus, antioxidant systems are activated during infection both in the host and the microbe to control metal toxicity. The importance of copper in bacterial plant infections has not been investigated, whereas the role of iron in the interactions of both animals and plants with their pathogens has been studied extensively. Iron is present in low amounts in plants, found mostly in the cell walls, vacuoles, plastids, and the nucleus[15,16]. In the plastids, iron is stored into proteins called ferritins, which play major roles as antioxidant[17]. The fight for iron between plants and bacteria has been thoroughly studied using the soft rot-causing bacterial pathogens *Dickeya* and *Pectobacterium*. The limited availability of iron in the plant apoplast leads to the *Dickeya dadantii* production of two siderophores to scavenge iron, achromobactin[18] and chrysobactin[19]. Both are required for the systemic progression of maceration symptoms in the host[20,21]. Lowering free iron levels by injection of the siderophores chrysobactin or deferrioxamine activates plant immunity in *Arabidopsis thaliana*[22,23]. In *A. thaliana* leaves, the accumulation of the AtFer1 ferritin around invading bacteria in response to *D. dadantii* infection is thought to avoid oxidative stress by scavenging iron and to deprive the bacteria from iron[12]. Iron is also a cofactor necessary for the activity of PelN, a pectate lyase secreted by *D. dadantii* required for full virulence on chicory leaves[24]. Interestingly, *A. thaliana* defensins AtPDF1.1 are antimicrobial peptides able to bind iron. Infection of leaves with *P. carotovorum* subsp. *carotovorum* induces the *AtPDF1.1* gene expression and an iron deficiency response in the plant[25]. Thus, iron plays multiple roles during plant bacterial infection and both partners try to control the metal availability.

In this study we show that IbpS, a protein secreted by *D. dadantii*, can bind $Fe^{3+}$ and $Cu^{2+}$. We demonstrate that exogenously added IbpS reduces the toxicity of $H_2O_2$, probably by preventing the Fenton reaction. Injection of this protein into *A. thaliana* leaves triggers an iron deficiency response. This protein has homologs in fungi and oomycetes, most likely owing to the result of horizontal transfer of a coding gene from bacterial origin. The reduced pathogenicities of *D. dadantii ibpS* and *Botrytis cinerea ibp* mutants reveal the importance of this conserved protein in the infectious process deployed by necrotrophic microbes probably because of their impaired capacity to detoxify ROS produced during infection. Our data suggest that Ibp proteins could represent a antioxidant protection mechanism common to necrotrophic phytopathogens.

## Results

**IbpS is a type II secretion system substrate**. The secretion of proteins by the type II secretion system (T2SS) is essential for the virulence of the soft rot bacteria *Pectobacterium atrosepticum* and *D. dadantii*[26]. Analysis of the *P. atrosepticum* SCRI1043 secretome identified several proteins secreted by the Out T2SS:[27] plant cell wall-degrading enzymes, the virulence factor Nep (NLP) and three proteins of unknown function, ECA2134, ECA3580, and ECA3946. We investigated whether homologs of these three proteins of unknown function exist and are secreted by *D. dadantii* 3937. A BLAST search[28] identified two proteins similar to ECA2134 in the *D. dadantii* genome: ABF-18996 (88% identity) and ABF-14625 (69% identity) (named hereafter as IbpS and IbpP, respectively, see below). These proteins possess a predicted signal sequence necessary for secretion by a T2SS and are annotated as periplasmic components of ABC transport systems. ABC transport systems are generally composed of one or two nucleotide binding proteins, one or two transmembrane proteins and a high-affinity periplasmic substrate-binding protein (SBP)[29]. Although genes coding for these proteins are usually organized in an operon in bacterial genomes, it is not the case for the genes encoding IbpS and IbpP. The *ibpS* gene is located in a cluster of genes encoding type three secretion system substrates (HrpK, HrpW, and DspE) and the contact-dependent growth inhibition protein CdiA (Supplementary Fig. 1). IbpP is positioned next to genes coding for the secreted proteases PrtA, PrtB, and PrtC and for their secretion system PrtF. In *P. atrosepticum*, ECA2134 is adjacent to the genes encoding the pectate lyase Pel, and KdgM1 and KdgM2, the two pectin degradation product outer membrane channels (Supplementary Fig. 1). The absence of genes encoding ABC transport components in the vicinity of these genes and the

presence of virulence factor genes in both bacteria support a role for these proteins in virulence rather than in transport. This hypothesis was also supported by transcriptomic analyses. In *A. thaliana* infection experiments by *D. dadantii*, *ibpS* was identified as one of the most-expressed genes at early stages[30,31]. We thus tested whether the presence of plant tissues could induce *ibpS* expression. When *D. dadantii* cells were cultured in the presence of chicory leaves (fresh or autoclaved), *ibpS* expression was increased by six-fold compared with the control (Supplementary Fig. 2). In contrast, pectin degradation products polygalacturonate and galacturonate, known to induce most *D. dadantii* virulence genes[32] had no effect on *ibpS* expression (Supplementary Fig. 2). The basal level of *ibpP* expression was very low and none of the tested conditions induced expression of this gene whose expression was undetectable in *D. dadantii* 3937.

We tested IbpS secretion in a "chicory-induced" *D. dadantii* culture using IbpS antibody. The protein could be detected in the extracellular medium of a wild-type *D. dadantii* strain culture but remained inside the cells in an *outD* mutant (Supplementary Fig. 3). IbpS is thus a T2SS substrate like ECA2134 in *P. atrosepticum*.

**ibpS has homologs in eukaryotes**. Proteins from the Ibp family were searched in the NCBI non-redundant database using the PSI-BLAST program and 898 proteins from 383 species/strains (222 bacteria and 162 eukaryotes; Supplementary Data 1) were retrieved. Ibp proteins are thus not only encoded by genes present in some Gram− and Gram+ bacteria, but also by genes in oomycetes, fungi and in two metazoa, the springtail *Folsomia candida* and the cereal cyst nematode *Heterodera avenae*. Interestingly, most of the organisms possessing at least one *ibp* gene are phytopathogens that have a hemibiotrophic or necrotrophic lifestyle (Supplementary Data 1). The genome of the hemibiotroph oomycete *Phytophthora sojae* harbors the most *ibp* genes (10) while BlastP searches on the EnsemblProtist website[33] showed that two other *Phytophthora* species (*P. parasitica* and *P. nicotianae*) and two necrotroph species (*Pythium ultimum* and *P. aphanidermatum*) also have at least six *ibp* genes (Supplementary Fig. 4). Putative signal sequences were predicted in Ibp proteins by the SignalP server[34] and also by the TatP 1.0 server[35] in the case of bacterial Ibp. Overall, putative signal sequences were predicted in 241 Ibp proteins from eukaryotes (over 301, 80%), 153 proteins from Gram+ bacteria (over 191, 80%) and 171 proteins from Gram− bacteria (over 405, 42%). The latter percentage is biased by an over-representation of *Pseudomonas* proteins in the NCBI non-redundant database. When the 231 proteins of *Pseudomonas* were excluded from the analysis, signal sequences were predicted in 143 Ibp proteins (over 174, 82%) from Gram− bacteria. Numerous Ibp proteins are thus potentially secreted.

To avoid species representation bias, 70 species were selected to study the evolution of their proteins belonging to the Ibp family (Supplementary Data 2). Eukaryotic Ibp proteins do not form a monophyletic group (Fig. 1), which, together with their patchy taxonomic distribution, suggests that they originate from distinct horizontal gene transfers (HGTs) from bacteria. The Ibp proteins of oomycetes and metazoa formed well-supported groups with bacterial proteins, among which *D. dadantii* IbpS and IbpP were found, in the trees constructed using two phylogenetic approaches (maximum likelihood (ML) and Bayesian). The fungal proteins of the Ibp family mainly form two well-supported classes that group together. Class I members were found in Basidiomycota species and some Ascomycota species, including *Botrytis cinerea*, whereas class II members were found in only Ascomycota species. In these two classes, the genes had

intronic regions but introns from the class I genes had no homology with those of the class II genes (Supplementary Fig. 5), which was the outcome expected if two independent HGTs led to these two classes. Other fungal proteins from the Zoopagomycota were grouped by only species. Overall, the evolutionary relationships of fungal proteins with the rest of the Ibp protein family were not clearly resolved in the phylogenetic topologies, providing no clue to the bacteria at the origin of the transfers to fungi. Finally, the different bacteria did not form monophyletic groups per taxonomic class. Instead, several well-supported nodes grouped proteins from different bacterial groups together suggesting that HGTs also occurred between bacteria.

**Ibp family members are metal-binding proteins**. To determine whether the functionalities of the proteins were conserved despite the genetic transfers and the phylogenetic distances, we investigated the properties and roles of the bacterial IbpS and IbpP proteins as well as those of the distant fungal *B. cinerea* BcIbp protein. Given that IbpS and IbpP in *D. dadantii* showed homology to SfuA, the ferric iron-binding protein of the SfuABC transporter, we speculated that Ibp proteins could also bind iron. When purified IbpS was incubated with Ni-NTA, Cu-NTA, Zn-NTA, and Fe-NTA resins, IbpS interacted with Fe-NTA and Cu-NTA matrixes but did not with Ni-NTA and Zn-NTA matrixes (Fig. 2a). Similarly, IbpP and BcIbp also bound the Fe-NTA and Cu-NTA resins (Supplementary Fig. 6).

To gain insight into the metal-binding property of the protein, metallation of IbpS by $Fe^{3+}$ was monitored. IbpS possesses 11 tryptophan residues in its mature form and can generate a strong fluorescence emission signal at an excitation wavelength $\lambda = 280$ nm. Addition of increasing amounts of $Fe^{3+}$ ions gradually quenched this signal demonstrating that this metal altered the environment of at least one tryptophan residue (Supplementary Fig. 7a). The binding curve was biphasic. A saturation was observed for one equivalent of Fe (Supplementary Fig. 7b). Then, the addition of increasing amounts of $FeCl_3$ did not lead to the saturation of the spectra (Fig. 2b). These results indicate that IbpS binds one Fe, and then that non-specific binding also occurs. Similar results were observed with IbpP. In these conditions it was not possible to determine an affinity constant from the spectra for the first site. Thus, an alternative approach, isothermal titration calorimetry (ITC) was utilized. However, the quality of the binding isotherm did not allow the determination of a $K_D$ likely owing to non-specific binding of Fe at high concentrations.

IbpS was retained on the Cu-NTA resin indicating that this protein can bind $Cu^{2+}$ ions. Fluorescence spectroscopy showed altered IbpS spectra in the presence of $Cu^{2+}$. Saturation of the titration was obtained for one equivalent of Cu (Fig. 2b). In order to determine the affinity, constant ITC experiments were performed. The titration of $Cu^{2+}$ into a solution of purified IbpS in 10 mM Tris-HCl buffer pH 7.0 induced significant enthalpy changes. A one-site binding model with stoichiometry of 1 was used to fit the data, yielding a Kd of 11 μM (Fig. 2c). These results show that IbpS binds one $Cu^{2+}$ with a micromolar affinity.

Taken together these results clearly indicated that IbpS and IbpP bound $Fe^{3+}$ and $Cu^{2+}$, which is the reason why ABF-18996 was named IbpS (for iron-binding protein secreted) and ABF-14625 IbpP, respectively.

**IbpS is prototypal of a novel class of substrate-binding proteins**. The crystal structure of the translocated region of *D. dadantii* IbpS (residues 28–372) was solved at a resolution of 1.7 Å using the single anomalous dispersion method (Table 1). The asymmetric unit of the crystal contains four chains A, B, C, and D that are nearly identical (rmsd < 0.4 Å). The structure of

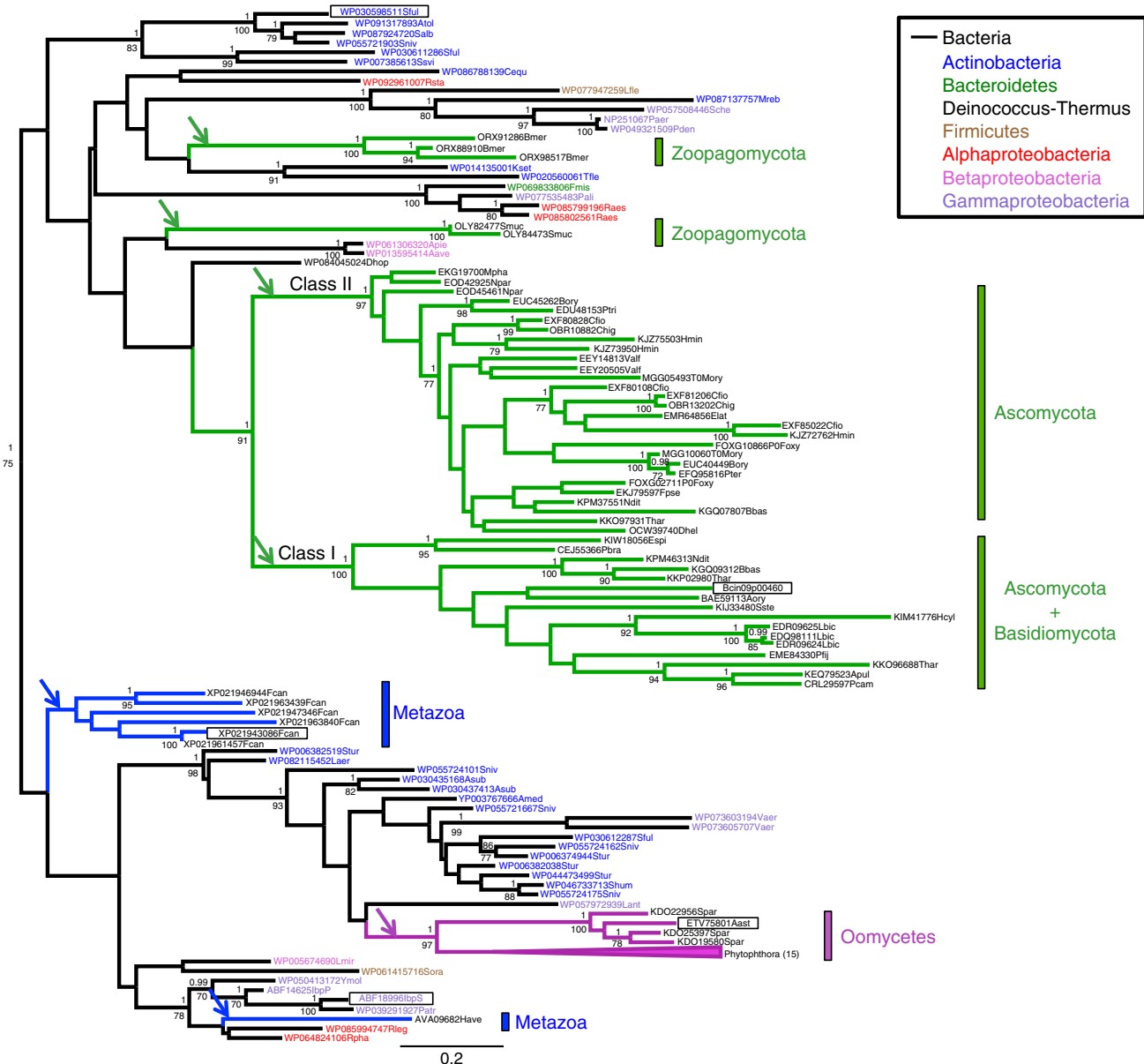

**Fig. 1** Phylogeny of IbpS and homologs. An alignment of 198 amino acids from 122 sequences was used to construct the phylogenies. The maximum likelihood tree was used as a reference topology. Bootstraps of interest ≥ 70 are shown below the branches and the corresponding posterior probabilities in the Bayesian topology are reported above. Arrows represent probable transfer events from bacteria to eukaryotes. The names of the proteins analyzed in this study are indicated in bold. The structure-based sequence alignment of the proteins with a boxed name is shown in Supplementary Fig. 8. The branches leading to fungal proteins are colored green, those leading to oomycete proteins are colored in red and those leading to metazoan proteins are colored in blue. The bacterial protein names are colored as described in the box. The corresponding species and sequence accession numbers are available in Supplementary Data 2

the IbpS monomer displays a typical SBP fold with two α/β lobes connected by a central hinge region consisting of an extended two-strand β-sheet (Fig. 3a and Supplementary Fig. 8). The SBP fold is associated with ligand binding to a pocket located between the two lobes via the so-called "Venus's flytrap"[36]. In absence of ligand the two lobes are in flexible open conformation and close upon substrate-binding. A structural homology search using DALI server[37] showed that IbpS is most similar of the SBP class II/cluster D[29] (Z score > 20), a subgroup that encompasses proteins interacting with a large variety of substrates such as carbohydrates, polyamine, tetrahedral oxyanions, or ferrous or ferric iron[29,38]. However, the structure of IbpS exhibits some unique features. The protein displays an additional α-helix at its N terminus and an extended C-terminal tail (residues 352–372) that,

to our knowledge, are not observed in the other members of the class II/cluster D family. The C-terminal tail connects the two lobes on the edge of the substrate-binding pocket (Fig. 3a) and appears to stabilize the protein in a conformation similar to a closed state, even in the absence of ligand. Furthermore, the central α-helix in the substrate-binding pocket, which is involved in iron or ligand binding in several class II/cluster D members such as AfuA[39] or FutA[40] is absent in *D. dadantii* IbpS structure and replaced by an extended loop (Fig. 3b). Consequently, the *D. dadantii* IbpS ligand-binding pocket is significantly different from that of other class II/cluster D family members and IbpS does not have the residues involved in iron coordination in FutA (four tyrosines and one histidine) (PDB code 3F11, Fig. 3b and Supplementary Fig. 8). Moreover the ligand-binding pocket is partly

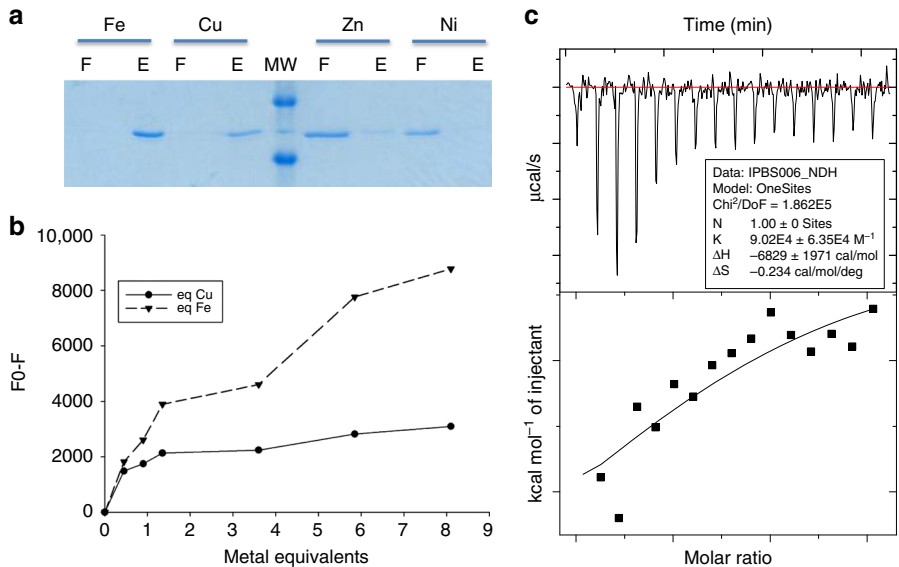

**Fig. 2** IbpS is a metal-binding protein. **a** IbpS (10 μg) in 100 μl of A buffer was incubated with 100 μl of Fe-NTA, Cu-NTA, Ni-NTA, or Zn-NTA resins for 15 min. After centrifugation the supernatants (flowthroughs) were removed and the resins were washed three times with 1 ml of buffer A. The protein was eluted with 100 μl of 50 mM EDTA and 10 μl of flowthrough (F) or eluate (E) was loaded onto an SDS-PAGE gel. **b** Fluorescence spectra of 70 μM IbpS titrated with the indicated equivalents of $FeCl_3$ (dashed line) or $CuCl_2$ (solid line). The F0-F values are plotted. **c** Isothermal titration calorimetry of IbpS. Cu (II)$Cl_2$ (1 mM) titrated into apo-IbpS (120 μM) in 10 mM Tris-HCl buffer (pH 7.0) at 30 °C. A blank run consisting of $CuCl_2$ dilution in the experimental buffer was performed and values subtracted from the assay. Top, raw data. Bottom, plot of integrated heats versus the Cu/IbpS ratio. The solid line represents the best fit for a one-site binding model

**Table 1 Data collection and refinement statistics**

|  | IbpS | IbpS-Iron | IbpS-SeMet |
|---|---|---|---|
| Data collection |  |  |  |
| Space group | P 1 21 1 | P1 | P 1 21 1 |
| Cell dimensions |  |  |  |
| $a, b, c$ (Å) | 77.45 | 58.89 | 76.88 |
|  | 112.31 78.72 | 78.89 83.91 | 112.26 79 |
| $\alpha, \beta, \gamma$ (°) | 90 95.53 90 | 73.25 | 90 94.8 90 |
|  |  | 74.74 84.92 |  |
| Wavelength | 0.9730 | 1.5895 | 0.9788 |
| Resolution (Å) | 47.55-1.7 | 47.99-1.8 | 47.72-1.80 |
|  | (1.761-1.7) | (1.86-1.80) | (1.89-1.80) |
| $R_{merge}$ | 0.06 (0.6) | 0.06 (0.5) | 0.09 (0.4) |
| $I /\sigma I$ | 8.6 (1.3) | 14.9 (2.5) | 16.3 (5.3) |
| Completeness (%) | 99.7 (99.4) | 86.3 (82.8) | 98.40 (89.9) |
| Redundancy | 3.4 (3.3) | 4.0 (4.0) | 14.2 (13.5) |
| Refinement |  |  |  |
| Resolution (Å) | 47.55-1.7 | 47.99-1.8 |  |
|  | (1.76-1.7) | (1.86-1.80) |  |
| No. of reflections | 146,072 | 111,650 |  |
|  | (11977) | (10703) |  |
| $R_{work} /R_{free}$ | 0.174 (0.314)/ | 0.155 (0.219)/ |  |
|  | 0.206 (0.345) | 0.191 (0.262) |  |
| No. atoms |  |  |  |
| Protein | 10947 | 11010 |  |
| Ligand/ion | 18 | 16 |  |
| Water | 1132 | 1129 |  |
| B-factors |  |  |  |
| Protein | 36.30 | 26.0 |  |
| Ligand/ion | 49.4 | 55.0 |  |
| Water | 43.2 | 34.8 |  |
| R.m.s. deviations |  |  |  |
| Bond lengths (Å) | 0.007 | 0.008 |  |
| Bond angles (°) | 1.03 | 1.11 |  |

*Values in parentheses are for highest-resolution shell

occluded by the C-terminal tail and the N-terminal part of α2. Instead, IbpS structure reveals a tunnel that runs perpendicular to the hinge β-strands and is open at both sides of the protein surface (Supplementary Fig. 9).

**IbpS forms dimers**. We observed two dimers in the crystal formed by A and D' (symmetry-related of chain D) and B and C' (symmetry-related of chain C) that were original and not found in other SBP structures to our knowledge (Fig. 4a). The interface buries 1230 Å² and is made of electrostatic interactions between α3 from chain A and α15 from chain D' and vice versa and between the two α15 helices. There, R337 and R344 make hydrogen bonds with the adjacent α15 E341. The interface also involves hydrophobic interactions between the two C-terminal tails that extensively interact with α15 and with each other. Notably, E353 inserts into the adjacent subunit to form a salt bridge with the buried R223 from the β6-α10 loops (Fig. 4b). Several residues involved in the dimerization, including R223 and R337 are strongly conserved (Supplementary Fig. 8), suggesting that dimerization might be a general feature of the Ibp protein family. To determine whether IbpS forms dimers in solution we performed size exclusion chromatography coupled with multi-angle laser light scattering (MALS) measurements. As seen in Fig. 4c, the protein eluted as a single peak with a calculated mass of 79 kDa, thus in agreement with a dimeric form (theoretical mass of 79 kDa). Because no other significant interface was found in the crystal, the A/D' dimer very likely exists in solution.

**A low-affinity metal-binding site is present in IbpS**. Given that IbpS bound iron and copper in our metallation experiments, we tried to co-crystallize the protein in the presence of various metals. Although crystals could be obtained in the presence of Zn, Mn, Cu, none of the crystal structures solved showed any evidence of metal binding. In the case of iron, the protein was co-crystallized with $FeCl_3$ and the structure of iron bound IbpS was

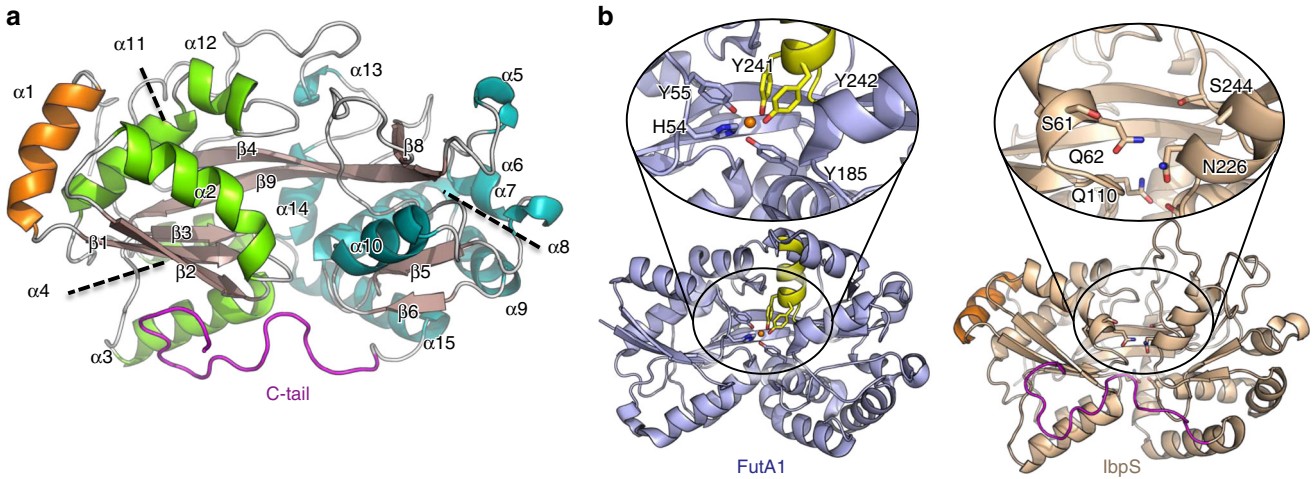

**Fig. 3** Structure of IbpS. **a** Ribbon representation of the crystal structure of IbpS (chain A). β-strands are colored in brown, α-helices are colored in green (N-lobe) or blue (C-lobe) except for α1 helix colored in orange. The C-terminal tail is colored in magenta. **b** Comparison of the iron-binding site of FutA1 (PDB code 3F11) with the corresponding residues in IbpS after superimposing the two structures. The iron atom is depicted as an orange sphere in FutA1. Residues Y241, Y242, Y55, Y185, and H54 coordinating iron in FutA1 are not conserved in IbpS. In particular the Y241/Y242-containing helix of FutA1 is replaced by a loop in IbpS, that has only a serine (S244) pointing towards the substrate-binding pocket

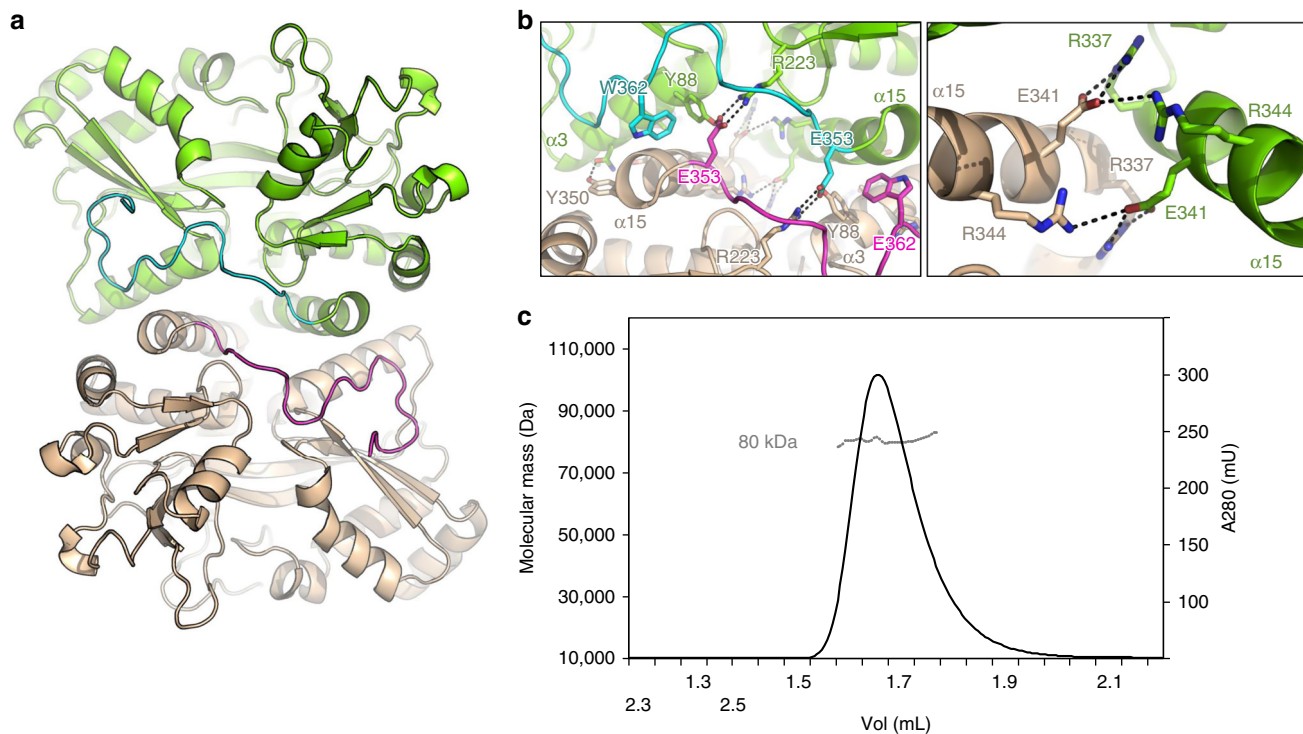

**Fig. 4** IbpS is a dimer. **a** Cartoon representation of the IbpS dimer between chains A (green) and D' (wheat). The C-terminal tails of chains A and D' are colored blue and magenta, respectively. **b** Close-up views of the dimer interface. The residues involved are indicated, and the side chains are shown as ball-and-sticks and colored like that in **a**. **c** SEC-MALS analysis of the IbpS dimer

solved at a resolution of 1.8 Å (Table 1). The protein crystalized in a different crystal form with the P1 space group and four molecules per asymmetric unit. All four molecules were bound to a single-iron atom at the same binding site (Fig. 5a–c). The structures of IbpS and Fe-IbpS were nearly identical and no important structural modification was observed. As suspected from our previous analysis, the metal did not bind at the canonical SBP cluster D protein substrate-binding pocket. Instead, the iron atom was located near the dimer interface (Fig. 5a), at the entry of tunnel identified previously (Supplementary Fig. 9). The metal sits in a cavity formed by residues S86, K87, Q110, T111,

M334, and a triple aspartate motif of the α5-β8 loop (D191, D192, D193, Fig. 5c) and containing a dense network of water molecules. In this negatively charged cavity the iron is poorly coordinated (only a water molecule according to the "Check my metal" server—https://csgid.org/csgid/metal_sites[41]) (Fig. 5c). This poor coordination might explain the weak binding affinity of IbpS observed in our metallation assays and suggests that metal binding is reversible. It is also possible that high-affinity binding to iron might require IbpS to bind a synergistic anion as described for *Neisseria gonorrhoeae* Ferric iron-binding protein[42]. Strikingly, when residues conservation based on the alignment of

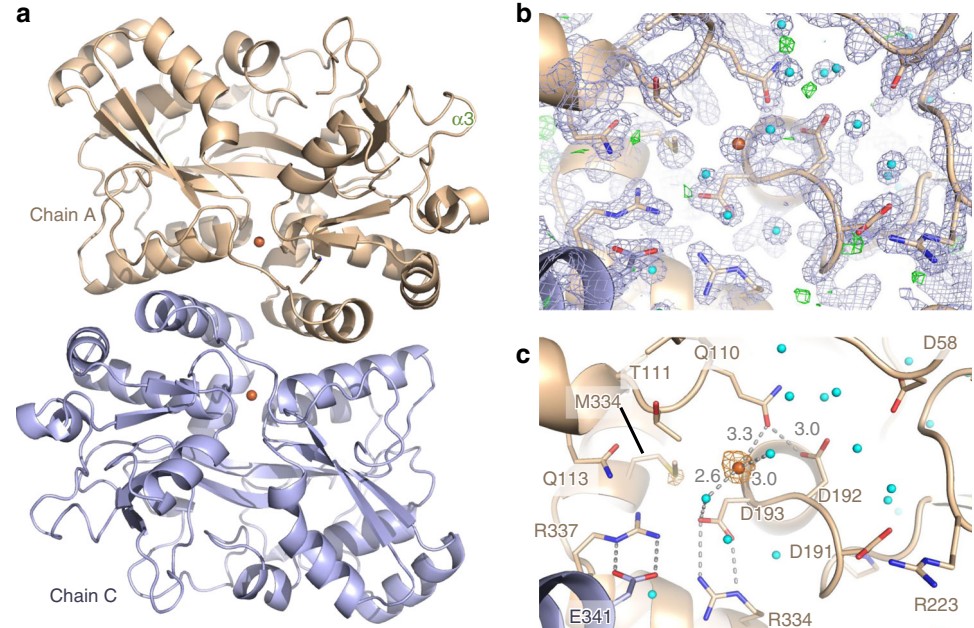

**Fig. 5** Structure of Fe-IbpS. **a** Cartoon representation of Fe-IbpS dimer with iron atom (orange sphere) bound to chain A (wheat) and C (light blue). **b** Close-up view of the iron-binding site in chain A. The structure is depicted in ball-and-stick with water molecules and iron atom represented as spheres colored in cyan and orange with an overlay of $2F_o-F_c$ electron density (blue mesh, contoured at 2.0 σ) and $F_o-F_c$ map (green mesh, contoured at 3σ). **c** Same view as **b** with an overlay of the anomalous electron density map contoured at 4σ (orange mesh). Interactions at the iron-binding sites are shown as gray dashed lines and distances are indicated in Å

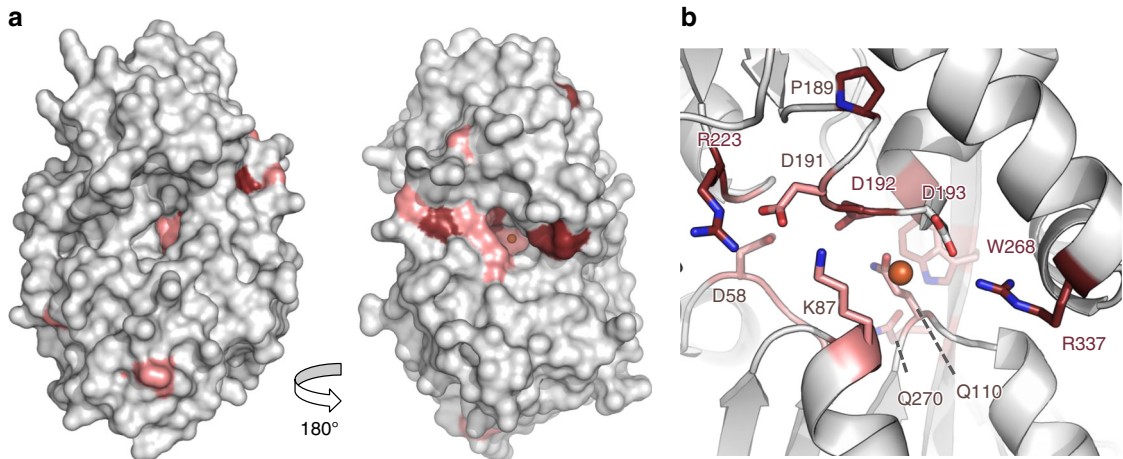

**Fig. 6** IbpS unusual iron-binding site involves conserved residues. **a** Two different orientations of a surface representation of the iron-binding site of IbpS. Strictly conserved residues are colored in firebrick and strongly conserved (95%) residues are colored in salmon (see also Supplementary Fig. 8). Sequence conservation was obtained from the alignment of 122 sequences used to generate the phylogenic tree (Fig.1). **b** Detailed view of the metal binding site with side chains shown as ball-and-stick. Iron atom is depicted as an orange sphere

122 sequences of IbpS homologs was mapped on the IbpS structure, most of the conserved residues (100% and 95% sequence identity) cluster at the iron-binding site, suggesting that this feature is critical for the protein function and conserved across the Ibp family (Fig. 6).

**Intracellular metal concentration modification by IbpS.** Metal-binding by a secreted protein could decrease the metal available in the external medium thereby decreasing the intracellular bacterial metal level. To test this hypothesis, we used the *acsA* gene expression as a sensor of intracellular $Fe^{2+}$ level. *acsA* encodes a protein involved in achromobactin synthesis and its expression is under the control of the transcription factor Fur in response to

$Fe^{2+}$ concentrations[21]. When bacteria were grown in an iron-depleted medium, *acsA* was expressed at a high level and the addition of 2 μM of $Fe^{3+}$ to the medium repressed its expression by 1.5-fold (Fig. 7a). Addition of $Fe^{3+}$ and 2 μM ethylene-*N*, *N'*-bis (2-hydroxyphenyl-acetic acid (EDDHA) to the medium prevented repression, as EDDHA chelates iron. The addition of 20 μM IbpS to the growth medium led to results similar to those obtained with EDDHA, suggesting that IbpS in the external medium reduced the intracellular iron concentration (Fig. 7a).

The intracellular $Cu^{2+}$ concentration was then monitored in *Escherichia coli* using the *luxCDABE* reporter gene fused to the autoregulated copper binding response regulator *czcR3* as a biosensor of intracellular $Cu^{2+}$ levels. The activity of the reporter increased as a function of external $Cu^{2+}$ concentration (Fig. 7b).

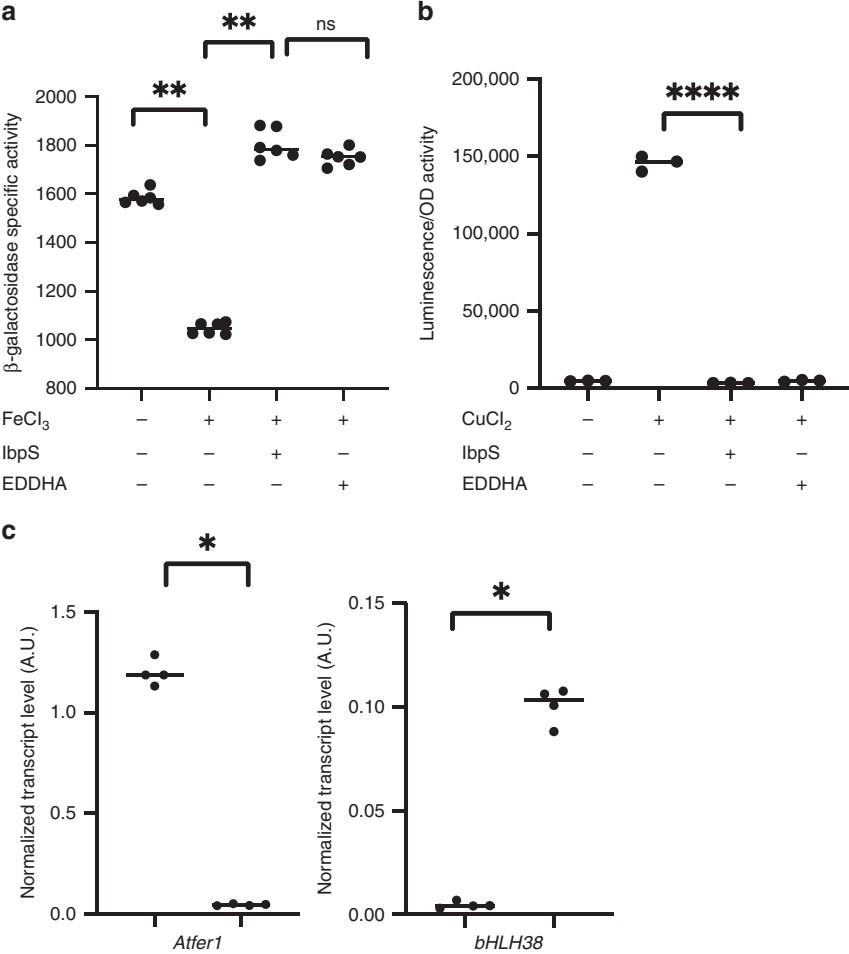

**Fig. 7** Modification of the intracellular metal concentration by extracellularly added IbpS. **a** *D. dadantii* strain A6050 containing the *acsA-lacZ* fusion was grown overnight in low phosphate + glycerol medium either supplemented or not with 2 μM FeCl₃, 20 μM IbpS, or 2 μM EDDHA. β-galactosidase activity was measured with *o*-nitrophenyl-β-ᴅ-galactose. The activities are expressed in μmoles of *o*-nitrophenol produced per minute and per milligram of bacterial dry weight + standard deviation. Data are expressed as the mean ($n = 6$) from six independent experiments. ns indicates non significative differences. ** denotes significant differences; $p < 0.005$ two-sided Mann and Whitney test. **b** *E. coli* strain W3110 containing the plasmidic *czcR3-luxCDABE* fusion was grown in low phosphate + glycerol medium either supplemented or not with 2 μM CuCl₂, 20 μM IbpS, or 2 μM EDDHA. Luminescence was recorded during 8 h; the value of the maximum activity recorded during the kinetics is plotted on the graph (luminescence (A.U.)/OD 600 nm). Data are expressed as the mean ($n = 3$) from three independent experiments. **** denotes significant differences; $p < 0.0005$, two-sided Mann and Whitney test. **c** Expression of the *A. thaliana Atfer1* and *bHLH38* genes. *A. thaliana* leaves were infiltrated with Buffer A or 6 μM IbpS in buffer A. After 24 h, RNA was extracted, reverse transcribed and subjected to real-time qPCR using gene-specific primers. Gene expression is indicated in arbitrary units and was normalized against synthetic constitutive gene *clathrin* and *actin* transcript levels. Experiments were performed four times with similar results. For each point, six plants were used, and three leaves per plant were infiltrated. Significant differences between control and IbpS treaments is indicated by *; $p < 0.05$ two-sided Mann and Whitney test. Source data are provided as a Source Data file

When IbpS was added exogenously together with Cu²⁺, the activity of the fusion dropped to a level comparable to that in the condition in which no Cu²⁺ was added (Fig. 7b). These results show that IbpS is capable of reducing the intracellular Cu²⁺ concentration by chelating it in the external medium.

To determine whether IbpS can affect the iron status *in planta*, leaves of *A. thaliana* seedlings were infiltrated with purified IbpS at a concentration of 6 μM. The expression levels of two plant genes known to respond to iron concentration were monitored. Expression of the gene encoding the plant transcription factor bHLH38 is upregulated under iron deficiency[43], whereas the ferritin-encoding gene, *AtFER1*, is upregulated under iron excess and repressed under iron deficiency[44]. Twenty-four hours after IbpS treatment, *bhlh38* expression was upregulated and *AtFER1* expression was decreased (Fig. 7c). These data indicate that the plant experienced iron deficiency stress upon IbpS treatment, which is consistent with the capacity of the protein to bind iron.

**Protective effect of Ibp proteins on bacteria exposed to H₂O₂.** Plant defense reactions involve the production of H₂O₂, which generates highly reactive ROS, such as OH°, that kill the pathogens in the presence of iron or copper. We hypothesized that IbpS secretion would reduce the concentrations of free iron and copper in the plant apoplasm, consequentially also reducing the production of ROS. In *Erwinia amylovora*, production of the siderophore desferrioxamine, which binds iron protects the bacteria from H₂O₂-induced death[45]. When *D. dadantii* cells were exposed to 5 mM H₂O₂ a strong decrease in viability was observed (Fig. 8a). When IbpS (0.5 or 5 μM) was added to the medium 5 min prior to the addition of H₂O₂, a protective ability against H₂O₂-induced death was observed. The same level of protection was obtained by adding the iron chelator EDDHA at 5 μM. A similar protection effect of IbpS was observed for *E. coli* cells exposed to 20 mM H₂O₂ (Fig. 8b). Thus, the binding of trace amounts of metal in the medium by IbpS limits H₂O₂-induced

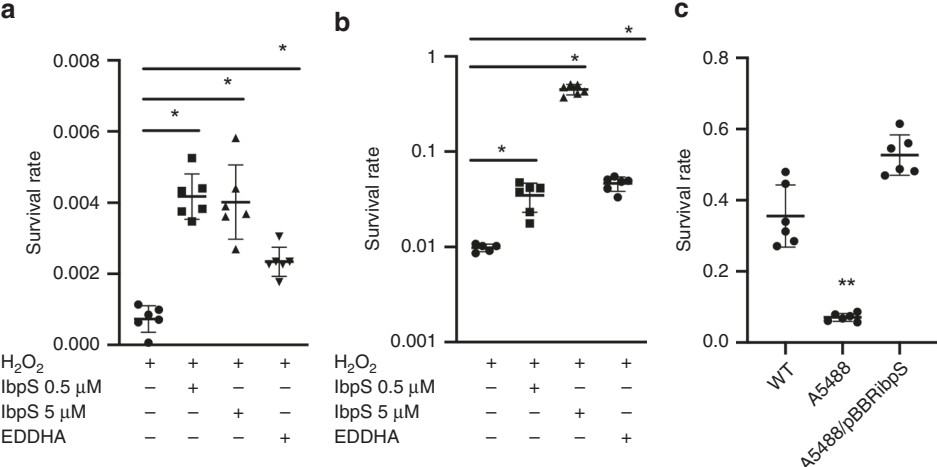

**Fig. 8** IbpS protects bacteria exposed to hydrogen peroxide. **a** *D. dadantii* A4922 or **b** *E. coli* NM522 cells were resuspended in water either supplemented or not with 0.5 or 5 μM IbpS or 5 μM EDDHA. After 5 min, 5 mM (**a**) or 20 mM (**b**) $H_2O_2$ was added. After 30 min, the number of viable bacteria was determined by plating serial dilutions on LB agar plates. **c** The *D. dadantii* A4922 wild-type strain (WT), the *ibpS* mutant and the complemented *ibpS/* pBBR*ibpS* strain were grown overnight in Tris medium plus glycerol containing chicory pieces to an OD = 1 and treated in the conditioned medium with 20 mM $H_2O_2$ for 1 hour. Viable bacteria number was determined by plating serial dilutions on LB agar plates. Mean (*n* = 6) and SD are represented on the dot plot graph. * denotes significant differences; $p < 0.005$, two-sided Mann and Whitney test. Source data are provided as a Source Data file

bacterial death. To test the effect of IbpS in a more physiological condition, the wild-type, the *ibpS* mutant and the complemented *ibpS*/pBBR*ibpS* strains were grown in Tris medium containing chicory pieces to induce IbpS synthesis (Supplementary Fig. 10). Bacteria in the conditioned medium were then exposed to $H_2O_2$ for 1 hour. As seen in Fig. 8c, an increased mortality rate was observed for the *ibpS* strain compared with the wild-type strain. This increased mortality was not observed for the *ibpS*/pBBR*ibpS* strain, demonstrating the direct effect of IbpS. We concluded that IbpS was able to protect *D. dadantii* cells from $H_2O_2$-induced death in a condition mimicking plant infection (Fig. 8c).

**Ibp proteins are involved in the infection process of necrotrophs**. In a large-scale transcriptome analysis of *B. cinerea* during cucumber infection, at 96 h post infection, the *Bcibp* gene (genome annotation: Bcin09g00460) was upregulated 12-fold[46]. We analyzed *Bcibp* gene expression during bean leaf infection: leaves were inoculated with a suspension of spores and *Bcibp* expression was detected as early as 16 h post infection (hpi) before emergence of symptoms. An increase in *Bcibp* expression was observed throughout the colonization and maceration of plant tissues (Supplementary Fig. 11). Thus *Bcibp* is expressed in *B. cinerea* and its expression is induced during plant infection.

*D. dadantii ibpS* and *B. cinerea* ΔBcibp strains were tested on various plants to estimate the role of the protein in virulence. A clear diminution of symptoms was visible on *A. thaliana* leaves infected with the *D. dadantii ibpS* strain compared with that in plants infected with the wild-type strain (Fig. 9a, b). The development of symptoms often stopped at stage 2 (part of the infected leaf maceration) and fewer completely macerated leaves and less generalization to the entire plant were observed. This phenotype was absent in the *ibpS*/pBBR*ibpS* strain (Supplementary Fig. 10, Supplementary Fig. 12). To test whether the production of IbpS protects bacteria from ROS produced by the plant, we performed infection tests with the *atrbohD-atrbohF A. thaliana* mutant, which is unable to produce $H_2O_2$ after a *D. dadantii* infection. No differences in the time of appearance or extension of symptoms were observed between the plants infected with the wild-type or with the *D. dadantii ibpS* mutant

(Fig. 9c, d). Thus, an IbpS-nonproducing strain is no longer disadvantaged when infecting a plant that does not produce $H_2O_2$. Although the *Arabidopsis* mutants may also be altered in other aspects of their immune responses, this result suggests a protecting role of IbpS against these defenses during infection.

The infection of cucumbers and beans with the *B. cinerea* ΔBcibp strain also revealed a defect in virulence (Fig. 9e, f). Six days after contamination with the mutant, 55% of the cucumber cotyledons and 43% of primary bean leaves exhibited no symptoms in contrast to the leaves infected with the wild-type strain which were totally macerated. In some cases (8% of cucumbers and 33% of beans) primary lesions were observed but did not progress (Fig. 9e, f). Thus, the absence of Ibp protein production led to the reduced virulence of both a bacterium and a fungus.

## Discussion

In this study, we have identified and characterized a family of proteins secreted by several plant pathogenic microorganisms. Our work establishes that members of the Ibp family exhibit the same fold as SBP proteins albeit with significant modifications making of IbpS structure a prototype for this family[29,38]. We found that IbpS is secreted in the outer medium by *D. dadantii* (Supplementary Fig 3). Given that its homolog in *Streptomyces scabies* is also secreted[47] and that many of the homologs identified herein have a signal sequence, these data collectively suggest that most Ibp are secreted proteins.

Ibp-encoding genes appear to have been transferred many times not only from prokaryotes to prokaryotes but also from prokaryotes to eukaryotes. These bacteria-to-eukaryote transfers, the most common inter-kingdom HGT event[48] are classified as 'maintenance transfers' when they originate from endosymbiotic organelles and as 'innovation transfers" when they provide the recipient with an additional functionality. Herein, we report an example, of an 'innovation transfer', with several independent transfers of bacterial Ibp-encoding genes to oomycetes, animals, and fungi. The antioxidant property displayed by this secreted metal-scavenging protein might provide protection to the recipients. In oomycetes, the transfer occurred prior to the radiation

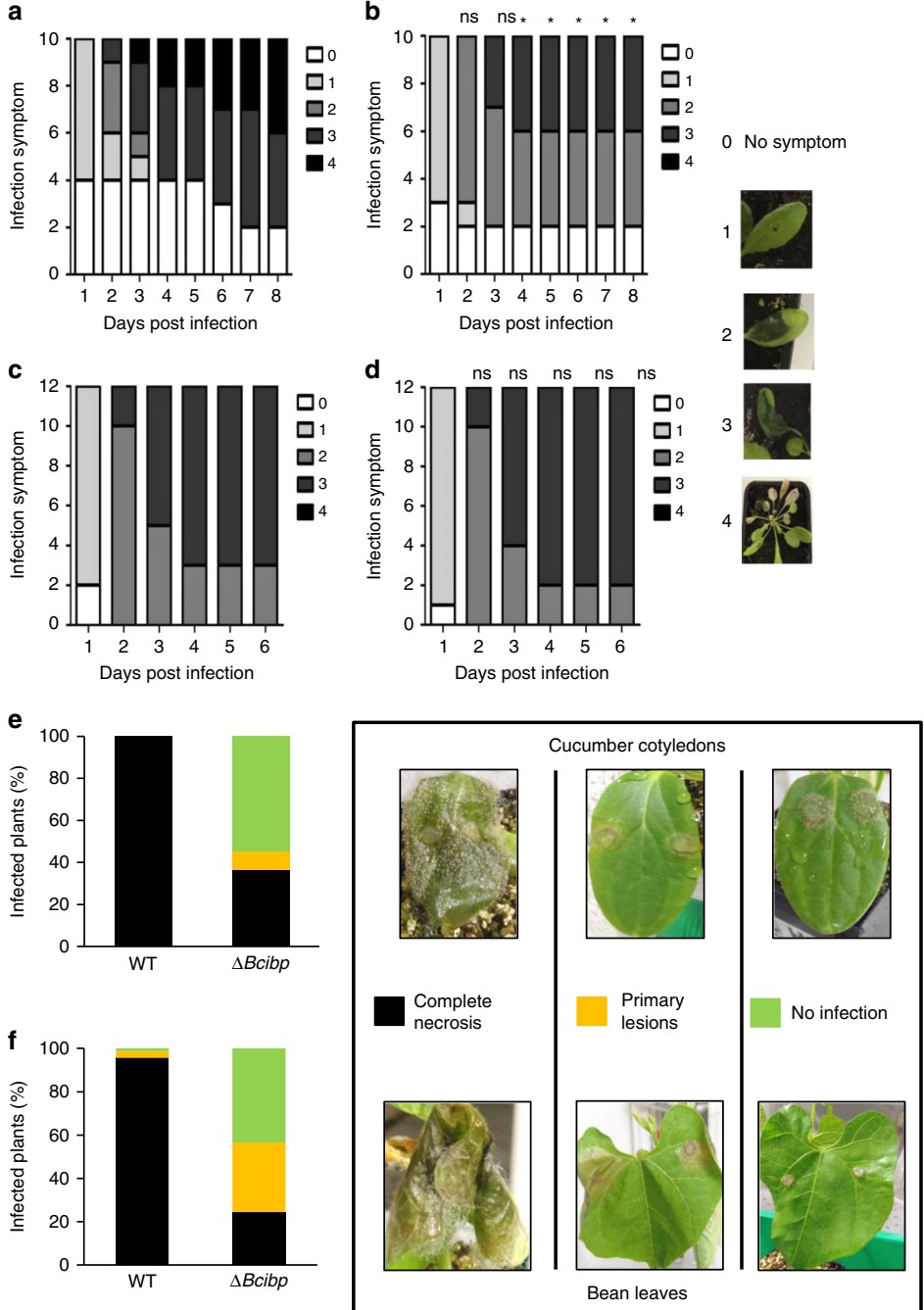

**Fig. 9** Reduced virulence of *D. dadantii* and *B. cinerea ibp* mutants. **a–d** Evolution of the symptoms on *A. thaliana* Col0 (**a**, **b**) and *atrbohD-atrbohF* (**c**, **d**) after inoculation with the wild type (**a**, **c**) or the *ibpS* mutant strain (**b**, **d**). Infection was performed on a single leaf by the deposition of a drop containing ~50 bacteria on a wound made by a needle. Symptoms were classified in five stages as shown on the right. Stage 0: no symptoms; stage 1, symptoms around the spot of infection: stage 2: maceration of the leaf limb; stage 3: maceration of the whole leaf, including the petiole; stage 4: generalization to the whole plant. * indicates a statistical difference (*p* < 0.05) between results of the wild type and the mutant strain at a given day (Khi2 likelihood ratio test). ns: not significant. No difference in class distribution was observed for the *atrbohD-atrbohF* mutant infection. One representative experiment out of four (for the Col0 infection) and out of three (for the *atrbohD-atrbohF* mutant infection) is represented. **e**, **f** Pathogenicity assays of the *B. cinerea* wild-type strain and Δ*Bcibp* strains on cucumber cotyledons (**e**) and French bean leaves (**f**). Seven-day-old plants were infected with 3-day-old mycelial plugs. Disease symptoms were scored 6 days after infection. The disease categories and symptom developments on both plant hosts are indicated on the right. All tests were performed in three independent experiments with at least fifteen leaves or cotyledons per strain for each test and two points of infection per leaf or cotyledon

of the different lineages, as Peronopsorales and Saprolegniales species possess monophyletic homologs. Several duplications resulted in a multigenic family, especially abundant in *Phytophthora* and *Pythium* spp., corresponding to hemibiotrophic and necrotrophic plant pathogen species, respectively. In contrast,

the obligate biotrophs, *Hyaloperonospora arabidopsis* and *Albugo* spp, and the nonpathogenic saprotroph *Thraustotheca clavata*, have secondarily and independently lost the *ibp* gene (Supplementary Fig. 4). Thus, the secreted Ibp protein could be of particular importance for pathogenicity, predominantly plant

pathogenicity. Many other HGTs detected thus far in oomycetes have putative functions associated with plant pathogenicity[49]. Two bacteria-to-animal HGTs of the *ibp* gene have also occurred. First, in the cereal cyst nematode *Heterodera avenae*, the *ibp* gene was shown to encode a putative effector[50,51]. HGTs of bacterial origins in plant parasitic nematodes have participated in the evolution of their capacity to parasitize plants[52], which might be an ongoing process as the *H. avenae* Ibp protein is very similar to that of *Rhizobium* spp., suggesting a recent transfer in this nematode. Second, in the parthenogenetic springtail arthropod *F. candida*, several duplications followed the transfer of the *ibp* gene, which now exists in six copies. These genes constitute some of the 809 'foreign' genes in the *F. candida* genome, of which 40% have a bacterial origin[53]. This widespread soil arthropod interacts and feeds on various plant pathogens[54]. Finally, *ibp* genes might have been transferred to fungi via multiple events as fungal Ibp proteins do not form a monophyletic group, and the two classes observed in Dikarya genomes do not share introns (Fig. 1 and Supplementary Fig. 5). Given the patchy distribution of species for these two classes, secondarily losses and fungus-to-fungus transfers most likely occurred. Many Pezizomycotina species harboring *ibp* genes, such as *B. cinerea* or *Magnaporthe oryzae*, are necrotroph or hemibiotroph phytopathogens. However, the correlation between the number of *ibp* genes and lifestyle is less clear than that observed for oomycetes. The maximum number of *ibp* genes observed in a fungal genome was four, observed in *Colletotrichum fioriniae* responsible for anthracnose in a wide range of crops and wild plants worldwide. However, the endoparasitic nematode *Hirsutella minnesotensis* and the plant symbiotic *Laccaria bicolor* also possess three genes each.

Controlling intracellular metal concentration reduces toxicity and ROS production. In bacteria, this can be achieved by not only reducing metal uptake but also by effluxing or sequestering the metal[55]. For example, overproduction of the FetA FetB iron exporter in *E. coli* increased its resistance to oxidative stress[56]. Iron can also be buffered in the cytoplasm via storage proteins, such as the ferritin FntA in *E. coli*. Sequestration in the periplasm is another way to control the metal concentration. The *Salmonella enterica* CueP protein reduces $Cu^{2+}$ to $Cu^0$ in the periplasm, which lowers its cytoplasmic concentration[57]. Controlling the intracellular metal concentration via sequestration outside the bacteria has been documented much less often. The copper binding protein CopM and the iron-binding protein FutA2 of *Synechocystis* PCC6803 have been found outside the cell, a location suggested to contribute to intracellular metal homeostasis[58,59].

During early steps of *A. thaliana* infection by *D. dadantii*, *ibpS* expression is strongly induced[30], whereas the chrysobactin and achromobactin siderophores are not yet synthesized, indicating that iron concentration is not limiting for bacterial growth at this stage of infection. Iron is strongly associated with cell walls in *A. thaliana* leaves and part of it is stored in plant ferritins[15]. Maceration of the tissues by pectinases contributes to liberate iron. Our work suggests that, given its rather low affinity for metal, IbpS might buffer iron concentration in the macerated tissue by transiently incorporating it to prevent its entry into the bacterial cell. The low affinity might enable the use of iron later in the infection process, for instance, by siderophores which have a much higher affinity for iron and are produced later by the bacterium. It is also possible that IbpS binds to a different ligand in vivo and has a much higher affinity for metal-complexes or requires structural re-arrangements that have not been observed in our study. Nevertheless, the conservation of the residues involved in iron-binding in IbpS strongly suggests that the binding site identified here accomplishes a common, essential function in all members of this family. Although Ibp proteins are

mostly found in necrotrophic phytopathogenic microorganisms, some are also produced by nonpathogenic or nonplant-associated microorganisms. Therefore, the proteins in these organisms may play a more general role in metal homeostasis or stress resistance.

## Methods

**Bacterial strains and media**. *D. dadantii* and *E. coli* strains used in this study are described in Supplementary Table 1. *D. dadantii* and *E. coli* cells were grown at 30 °C and 37 °C, respectively, in LB medium or M63 minimal medium supplemented with a carbon source (2 g/l). When required antibiotics were added at the following concentration: ampicillin, 100 µg/l, kanamycin and chloramphenicol, 25 µg/l. To induce IbpS synthesis, 2 g of chicory leaf tissue were added in 5 ml of M63 medium. Media were solidified with 1.5 g/l agar.

**Fungal strains, growth conditions and phytopathogenicity assays**. *Botrytis cinerea* (teleomorph *Botryotinia fuckeliana* (de Bary) Whetzel) strain B05.10 was maintained on malt solid sporulation medium (malt extract 15 g/l, glucose 5 g/l, yeast extract 1 g/l, tryptone 1 g/l, acid hydrolysate of casein 1 g/l, ribonucleic acid 0.2 g/l and agar 20 g/l)[60], and was used as a recipient strain for genetic modifications. For DNA preparations, the mycelium was grown on solid sporulation medium for 3 days on the surface of cellophane membranes inoculated with 4-mm mycelial plugs. Infected plant tissues were collected after inoculation of 1-week-old French bean (*Phaseolus vulgaris var Saxa*) leaves using conidia collected from 10-day-old sporulation medium cultures. Ten 50 µl-droplets of a conidial suspension ($10^3$ conidia/ml in sporulation medium) were deposited on the surface of each leaf. Ten leaves were inoculated for each experimental condition. The infected plants were incubated at 21 °C under 100% relative humidity and dark (10 h)-daylight (14 h) conditions. At different stages of symptom development, leaves were harvested and frozen at −80 °C. Infection assays were also performed with cucumber (*Cucumis sativus*) cotyledons using 4 mm plugs collected from 3-day-old mycelium sporulation medium.

**Construction of *D. dadantii* and *B. cinerea* mutants**. *D. dadantii ibpS* mutant. A DNA fragment containing the *ibpS* gene was amplified with the primers 18996 L + and 18996L− (Supplementary Table 2) and ligated into plasmid pGEM-T (Promega). A *uidA*-kan cassette was inserted into the unique *Kpn*2I site to create a *ibpS*–*uidA* transcriptional fusion. The construction was introduced into *D. dadantii* by electroporation. After several cultures in low phosphate medium[61] in the presence of kanamycin, bacteria were plated on GL + kanamycin. Recombination and loss of the plasmid were checked by replica plating on GL + ampicillin. Correct recombination was checked by PCR using oligonucleotides flanking *ibpS*. *B. cinerea* Δ*Bcibp* mutant. Knockout mutants were constructed using a gene replacement strategy (Supplementary Fig. 13), and the deletion DNA cassette was generated by double-joint PCR. The 5'(1080 bp)- and 3'(522 bp)-flanking regions of *Bcibp* were amplified from *B. cinerea* genomic DNA (100 ng) using the primer pairs For-5'-ibp/Rev-5'-ibp and For-3'-ibp/Rev-3'-ibp, respectively (Supplementary Table 2). Rev-5'-ibp and For-3'-ibp also contained sequences homologous to the hygromycin resistance cassette containing the *hph* gene under control of the *trpC* promoter of *A. nidulans* and previously amplified from pFV8[62] using the primers For-Hygro and Rev-Hygro. Purified amplicons were combined and amplified together using the nested primers PCR3-FOR and PCR3-REV, which bind within the 5' upstream and the 3' downstream fragments of the target gene, respectively. The gene replacement cassette was verified by DNA sequencing. *B. cinerea* transformation was carried out using $2.10^7$ protoplasts transformed with 1.5 µg of DNA and plated on isotonic medium ($KH_2PO_4$ 0.2 g/l, $MgSO_4$ 0.1 g/l, KCl, 0.1 g FeSO_4,7H_2O 2 mg, saccharose 200 g, $NaNO_3$ 2 g and agar 15 g/l) supplemented with 70 µg/ml hygromycin (Invivogen, France) and incubated at 23 °C in the dark[63]. Transformants were picked after 5–10 days and cultured on minimal medium ($KH_2PO_4$, 1 g/l, $MgSO_4$ 0.5 g/l, KCl 0.5 g/l, $FeSO_4,7H_2O$ 1 mg/l, glucose 20 g/l, $NaNO_3$ 2 g, and agar 15 g/l) supplemented with 70 µg/ml hygromycin (Invivogen, France). Diagnostic PCR was performed to detect homologous recombination in the selected hygromycin-resistant transformants using the primer pairs P1/P2 and P3/P4.

**Protein purification, antibody production, and western blotting**. The coding sequences of *ibpS*, *ibpP*, and *Bcibp* without signal sequence were amplified with the primers 18996GEX+ and 18996GEX−, 14625GEX+ and 14625GEX−, and Bc96GEX+ and Bc96GEX−, respectively (Supplementary Table 2). The amplified DNAs were digested with *Bam*HI and *Xho*I and ligated into pGEX-6p3 plasmid (GE Healthcare) digested with the same enzymes. The pGEX derivatives producing the fusion proteins were introduced into *E. coli* NM522 strain. Cells were grown in LB medium to $OD_{600}$ 0.8 and protein production was induced with 1 mM isopropylthiogalactoside (IPTG) for 3 h. Cells were collected by centrifugation, resuspended in buffer A (50 mM Tris pH 7.0, 100 mM NaCl) and broken in a French cell press. Unbroken cells were eliminated by centrifugation. The fusion proteins were bound on Protino Glutathione Agarose 4B (Macherey-Nagel) equilibrated with buffer A, washed several times with the same buffer and the proteins were liberated by addition of Prescission protease (GE Healthcare) according to the manufacturer's protocol. The protein was incubated with 1 mM

ethylenediaminetetraacetic acid (EDTA) to remove any trace metal. EDTA was eliminated using PD-10 buffer exchange columns (GE Healthcare). The purified protein was injected to a rabbit for antibody production (Covalab, Villeurbanne, France). For western blot, proteins were separated by sodium dodecyl sulphate-polyacrylamide gel electrophoresis (SDS-PAGE) and transferred onto a poly-vinylidene difluoride membrane (Millipore). Anti-IbpS antibodies were used at a dilution 1/10,000. Unprocessed scans are provided in the Source Data file.

For crystallogenesis experiments, the protein was loaded on a Superdex 75 column equilibrated in 50 mM Tris-Cl pH 7.0, 100 mM NaCl. IbpS eluted as a single peak and fractions were pooled and concentrated to 10 mg/ml. Protein concentration was determined using a Nanodrop spectrophotometer (Thermo Scientific). Labeling of IbpS with selenomethionine was performed by growing $E.$ $coli$ NM522/pGEX-IbpS in M63 + glucose medium. When $OD_{600}$ reached 0.5, selenomethionine (40 μg/ml), 1 mM IPTG and all amino acids except methionine and cysteine were added at a concentration of 0.01%. Cells were grown overnight and treated as described for wild-type IbpS. Pure SeMet-IbpS was concentrated to 6.6 mg/ml. The presence of three seleno-methionines was confirmed by mass spectrometry using a Voyager-DE Pro MALDI-TOF mass spectrometer (Sciex) equipped with a nitrogen UV laser ($\lambda = 337$ nm, 3 ns pulse). The instrument was operated in the positive-linear mode (mass accuracy: 0.05%) with an accelerating potential of 20 kV. Typically, mass spectra were obtained by accumulation of 600 laser shots and processed using Data Explorer 4.0 software (Sciex). Samples were mixed with sinapinic acid (saturated solution in 30% acetonitrile and 0.3% trifluoroacetic acid), deposited on the MALDI target and air-dried before analysis. Spectra obtained for WT-IbpS and for SeMet-IbpS show, respectively, MH + at m/z 39493.3 and 39638.7 corresponding to the expected increment in mass when replacing the three Met by three SeMet.

**MALS**. Size exclusion chromatography (SEC) experiments coupled to MALS and refractometry were performed on a Superdex S200 5/150 GL increase column (GE Healthcare). 25 μl of IbpS protein were injected at a concentration of 10 mg ml⁻¹ in 50 mM Tris-Cl pH 7.0, 100 mM NaCl. On-line MALS measurement was carried out with a miniDAWN-TREOS detector (Wyatt Technology Corp., Santa Barbara, CA) using a laser emitting at 690 nm and by refractive index measurement using an Optilab T-rex system (Wyatt Technology Corp., Santa Barbara, CA). Weight averaged molar masses (Mw) were calculated using the ASTRA software (Wyatt Technology Corp., Santa Barbara, CA).

**Crystallization, structure determination, and refinement**. Crystallization conditions were screened at 19 °C by the sitting-drop vapor-diffusion method and commercial kits from Hampton Research, Molecular Dimensions Limited and Qiagen in 96-well plates (TTP Labtech iQ plates and Molecular Dimensions Limited MRC plates) with a TTP Labtech Mosquito nanodispenser. Initial crystals were obtained with IbpS at 10 mg/ml with a reservoir solution containing 30% PEG 4000, 80 mM magnesium acetate, 50 mM sodium cacodylate pH 6.5 (condition C1 of Natrix crystallization screen, Hampton Research) and 30% PEG 1500 (condition D7 of Crystal Screen crystallization screen, Hampton Research). Optimized crystals of IbpS grew in 24% PEG 4000, 80 mM magnesium acetate, 50 mM sodium caco-dylate pH 6. IbpS-SeMet crystals were obtained in 28% PEG 4000, 80 mM mag-nesium acetate, Tris-HCl 100 mM pH 7.4. Crystals were cryoprotected with the crystallization condition supplemented with 15% ethylene glycol and 10% glycerol for native crystals and SeMet crystals, respectively, and flash cooled in liquid nitrogen. To obtain the Fe-bound IbpS structure, concentrated protein was incubated with increasing concentrations of FeCl₃ to a final concentration of 1.8 mM. The solution was then filtrated on a Zeba column (Thermofisher) and drops were immediately set up. Best crystals were obtained in 20% PEG 3350, 0.1 M BisTrisPropane pH 8.5, 0.2 M NaNO₃ (condition H5 of PACT1er crystallization screen Molecular Dimensions). X-ray diffraction data from native and selenomethionine derivative crystals were collected at 100 K at the European Synchrotron Radiation Facility (in Grenoble, France) beamline ID23EH1 for both data. Crystals of native and SeMet-IbpS diffracted to a resolution of 1.7 Å and 1.8 Å, respectively. They belonged to the space group P2₁ and contained four molecules per asymmetric unit (Table 1). Crystals of Fe-bound IbpS diffracted to 1.8 Å and belonged to the space group P1 with four molecules per asymmetric unit. Diffraction data were indexed and integrated using XDS[64] and scaled with SCALA from the CCP4 program suite[65]. Data collection statistics are given in Table 1. The structure was solved by the single wavelength anomalous dispersion method with SeMet crystals data using the Autosol program of the PHENIX suite[66]. Phasing statistics are given in Table 1. The experimental map was of excellent quality and approximately 80% of the model could be built automatically with the Autobuild program and completed by manual building using COOT[67]. The model was used as a template for molecular replacement in the native dataset. The atomic positions and TLS parameters were refined using PHENIX[66] (Table 1) Structure refinement was performed using the Phenix program Refine[68] to a Rfree/Rwork of 0.17/0.20. A stereo view of the final 2F0-Fc density map is shown in Supplementary Fig. 14. The structure of Fe-IbpS was solved by molecular replacement and refined to a Rfree/Rwork of 0.15/0.19. Both structures have excellent geometry parameters with 98% of residues in the favored and 2% in the allowed regions of their respective Ramachandran plots.

**Analysis of gene expression in planta**. Five week-old seedlings of $Arabidopsis$ $thaliana$ Col0 were grown under short days (8-h light/16-h dark) with 21 °C temperature and 70% relative humidity. Leaves were infiltrated with the indicated treatments using a syringe without a needle, then harvested 24 h after treatment followed by freezing in liquid nitrogen. Total RNAs were purified with TRIzol reagent (Thermo Fisher Scientific) according to the manufacturer's instructions. Total RNA concentration was determined using a NanoDrop ND-1000. RNA samples were treated with DNaseI-RNase free (Thermo Fisher Scientific) to remove any DNA contamination. A total of 1 μg of DNase treated RNA was reverse transcribed using the RevertAid (Thermo Fisher Scientific) and 0.5 μg of oligodT primers following the supplier's instructions. One μl of the 1:10 diluted cDNA was subjected to real-time qPCR using SYBR Green PCR Mastermix (Takyon, Euro-gentec) and gene-specific primers ($bHLH38$ Forward, $bHLH38$ Reverse; $AtFER1$ Forward, $AtFER1$ Reverse; $Clathrin$ Forward, $Clathrin$ Reverse; PP2a3-F, PP2a3-R). The reference gene $clathrin$ and PP2a were used to normalize gene expression profiles.

**Analysis of gene expression in B. cinerea**. DNA-free total RNA was extracted from infected bean leaves[69] and 2.5 μg were used for the synthesis of cDNA using the Thermoscript RT-PCR system kit (Invitrogen, USA) according to the manu-facturer's recommendations. Real-time quantitative PCR experiments were per-formed in 96-well plates using ABI-7900 Applied Biosystems (Applied Biosystems, USA). Primer pairs were designed using Primer Express (Applied Biosystems, USA). The amplification reactions were carried out using SYBR Green PCR Master Mix (Applied Biosystems, USA) with the following protocol: 95 °C for 10 min and 40 cycles of 95 °C for 30 s and 60 °C for 1 min. Relative quantification was based on the $2^{-\Delta Ct}$ method using the actin-encoding gene (Bcin16g02020), the $bcef1\alpha$ gene (Bcin09g05760) and the $bcpda1$ gene (Bcin07g01890) as normalization internal controls. Three independent biological replicates were analyzed.

**Virulence tests on plants**. Infection of 6-week-old $A.$ $thaliana$ Col0 and the $atrbohD$-$atrbohF$ mutant[70] by $D.$ $dadantii$ was performed according to Lebeau et al.[71]. One leaf per plant was wounded with a needle and inoculated by depositing a 5 μl droplet of a bacterial suspension at 10⁴ bacteria/ml in water. Plants were kept at 22 °C at high humidity. The virulence was estimated daily by visual examination of leaf damage with the following scale: 0 = no symptoms, 1 = symptoms around the spot of infection, 2 = maceration of the leaf limb, 3 = maceration of the whole leaf, including the petiole, 4 = generalization to the whole plant. Virulence scores were not blinded.

**Iron-binding experiments**. In all, 100 μl of Ni-NTA, Zn-NTA, Cu-NTA, or Fe-NTA resins equilibrated in buffer A were incubated with 10 μg of protein in the same buffer for 15 min. After centrifugation the supernatant was withdrawn and the resins were washed three times with 1 ml of buffer A. The protein was then eluted with 100 μl of 50 mM EDTA and the fractions were analyzed by SDS-PAGE.

**Fluorescence spectroscopy**. Fluorescence quenching of IbpS was monitored between 300 and 400 nm with an excitation wavelength of 280 nm. Protein was dissolved in 50 mM Tris-HCl, 100 mM NaCl buffer (pH 7.0) to a final concentration of 70 μM. Metal ions solution was titrated stepwise into the protein solution with a volume variation below 1% and incubated for 5 min at room temperature before the analysis. Each addition was made from a 100-fold concentrated metal solution to give a final one-fold metal concentration. $Fe^{3+}$ ions were given by a solution of FeCl₃.

**H₂O₂ killing tests**. To test the effect of IbpS added exogenously, bacteria grown in LB medium overnight were centrifuged, resuspended in water at an $OD_{600} = 1$ and diluted 1000-fold in water. In total, 100 μl of this bacterial suspension were incu-bated for 5 min with 5 μM EDDHA or 0.5 or 5 μM IbpS. In all, 5 mM (for $D.$ $dadantii$) or 20 mM (for $E.$ $coli$) H₂O₂ were added. After 30 min, viable bacteria number was determined by plating serial dilutions on LB agar plates. To test the effect of IbpS produced at a physiological level by bacteria, the wild-type strain A4922, the $ibpS$ mutant strain and the complemented $ibpS$/pBBR$ibpS$ strain were grown in Tris medium[61] plus glycerol containing chicory pieces to an OD = 1. The conditioned medium containing the bacteria was treated with 20 mM H₂O₂ for 1 h. Viable bacteria number was determined by plating serial dilutions on LB agar plates.

**Enzymatic assays**. β-glucuronidase assays were performed on toluenized extracts of cells grown to exponential phase by the method of Bardonnet et al.[72] using $p$-nitrophenyl-β-D-glucuronate as the substrate. β-galactosidase assays were per-formed on toluenized extracts of cells grown to exponential phase with $o$-nitrophenyl-β-D-galactose as the substrate.

**Luminescence assays**. The assays were conducted in 96-well plates. The wells were filled with 200 μL of 0.4 % glucose M63 minimal media containing increasing concentrations of metals, inoculated with 10⁶ bacteria harvested at $OD_{600}$ of 0.6. The plate was sealed using gas-permeable Breathe Easy membrane (Sigma Aldrich)

and placed into a "TECAN Infinite Pro" plate reader equilibrated at 37 °C and programmed to measure $OD_{600}$ and luminescence every 20 min, after a 1-min period of shaking, during 12–20 h. Background OD and luminescence (values at time = 0) were subtracted to each data point. To calculate the maximal activity, for each well, the ratio (luminescence/OD) was plotted as a function of time, and the maximal value was conserved. By doing so, we took into account the eventual growth lag between two conditions.

**Analysis of the Ibp protein family**. The closest homologs of *ibpS* and *ibpP* were searched, using PSI-BLAST[28] with the *D. dadantii* (ABF-18996) protein sequence as query, in the NCBI non-redundant (nr) database. Four iterations were done, with a threshold of $1e^{-50}$, retrieving 897 protein sequences. This threshold was chosen because, at this level of the results, there was a relatively important increase in e-values, from $e^{-54}$ to $^{-48}$, suggesting that proteins with stronger *e* values do not correspond to Ibp proteins. PSI-BLAST hits, with e-value worse than the threshold, up to $1e^{-9}$, were displayed to see the proteins that were the most similar to the obtained Ibp family. These hits corresponded to hundreds of bacterial SBP, mainly from the D cluster of SPB proteins, but did not contain other eukaryotic similar sequences. The 897 proteins of the Ibp family were aligned using MAFFT with the L-INS-i algorithms and default settings[73]. Different ML phylogenetic trees were constructed to explore the sequence diversity, with the method detailed below for the final selected sequences, but with the approximate likelihood ratio test for branches[74] instead of bootstraps. These trees permitted the selection of a subset of 71 species representative of the diversity of the taxa, and which Ibp sequences reflected well the diversity observed among the 897 detected Ibp. For the analysis of Ibp proteins in these species, we kept the Ibp detected in the complete proteome of a unique selected strain, downloaded from an Ensembl site when available[33]. 122 sequences were thus selected and aligned with MAFFT and the L-INS-i algorithm[73]. Consensus sequences, allowing gaps, were constructed, using Seaview[75], to detect the residues strictly (100%) or strongly (≥95%) conserved.

Phylogenetic relationships between the selected Ibp proteins were inferred using ML and Bayesian approaches. In both cases, regions that were suitable for phylogenetic inference were selected from the multiple alignment, using BMGE with default parameters[76]. For the ML approach, ProtTest3 was used for determination of amino-acid sequence evolution best-fit model[77]. According to the Akaike information criterion corrected for small sample size, the best-fit model was the LG model[78] with rate variation among sites (+G). The ML tree was then constructed with PhyML 3.0[79] using the following parameters: the LG model, an estimated gamma distribution of rates of evolution, subtree pruning and regrafting and five random starting trees added to the standard BioNJ starting tree. The support of the data for each internal branch of the phylogeny was estimated using non-parametric bootstraps, with 100 replicates. For the Bayesian approach, MrBayes v3.2.6[80] was used with a mixture of models and an automatic detection of the model with highest probability as well as an estimation of the gamma distribution of the rates of evolution. The best scoring model of evolution was determined to be the WAG model with a probability of 1.0. The program was run with four chains for 1,000,000 generations and trees were sampled every 500 generations. To construct the consensus tree, 25% of the trees were eliminated following a burn-in process and posterior probabilities were used as support for internal branches. The resulting trees were edited with Figtree v.1.4.3.

**Statistical analysis**. To analyze plant infection symptoms, a contingency analysis (Khi2 likelihood ratio test) was performed on symptom classes everyday. The results presented in the Figures correspond to the mean values + standard deviation (error bars). The significance level analysis was performed with a Mann and Whiney test.

**Reporting summary**. Further information on research design is available in the Nature Research Reporting Summary linked to this article.

## Data availability

Atomic coordinates and structure factors have been deposited in the Protein Data Bank with accession code 6FJL for native IbpS and 6HHB for Fe-IbpS. Source data for Figs. 2a, 7, 8, and Supplementary Figs. 3, 6, 10, and 11 are provided with the paper.

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

## Acknowledgements

We thank Céline Vannoz and Florence Ruaudel for excellent technical work, members of the MAP laboratory and D. Expert for reading the manuscript. We are grateful to Elise Lacroix in charge of the greenhouse facilities of the FR BioEnvis at the University Claude Bernard, Lyon, for her assistance. We thank Dr. Didier Nurizzo and other members of the ESRF Staff for assistance in data collection and Roland Montserret for help with ITC experiments. We acknowledge the contribution of Protein Science Facility of SFR Biosciences (UMS3444/CNRS, US8/Inserm, ENS de Lyon, UCBL) for protein crystallization and mass spectrometry analysis.

## Author contributions

G.C. conceived and designed the study. V.G.-C and L.T. performed the crystallography experiments, I.G. analyzed the phylogeny, M.R. and A.D. analyzed the plant response, E.L., C.R., and N.P. performed *Botrytis* studies, L.L., A.R and G.C. performed biochemistry and *Dickeya* experiments. L.T, I.G, A.R, A.D., N.P. and G.C. wrote the article.

## Competing interests

The authors declare no competing interests.
