## [Peer Review File · Nature Communications]

Reviewers' comments:

Reviewer #1 (Remarks to the Author):

Title: Reactive oxygen species protection of necrotrophic phytopathogens by a conserved secreted metal binding protein.

This manuscript describes the structure of a secreted protein (IbpS) related to ABC transporter solute binding proteins and its function in sequestering reactive metals to ameliorate oxidative stress in plant pathogens. This is an interesting and, to my knowledge, novel finding and seems to be applicable to a wide range of species. Gene expression and virulence assays are convincing and indicate that IbpS does protect the organism from oxidative stress and is important for full virulence. However, there are major problems with the in vitro metal binding assays, which suggest a very weak binding affinity for both Fe and Cu that cannot be physiologically relevant. Since the Ibp proteins are proposed to exert their antioxidant functions by metal binding, this is a major point that needs to be addressed. The analysis of structural data is also somewhat lacking. Specific points with regard to these issues are indicated below along with a few minor issues.

In vitro metal binding assays:

- The major problem with all of the assays is that saturation is never demonstrated. If a spectroscopic assay is to be used, a plot of fluorescence/absorbance versus [metal] must be shown and it must saturate. Whatever assay is used, binding affinity and stoichiometry for both metals should be determined. Also, why is there no data for IbpP?
- ITC is probably the best way to quantify metal binding, but the data presented is of poor quality and only given for Cu. Here again, the titration is not run to saturation. Further, a blank run is not performed to indicate that the heat changes observed are not simply a result of dilution of the CuCl₂ solution.

Crystallographic analysis:

- The authors state that IbpS does not contain the 4 Tyr and His residues that bind iron in FutA. What does it contain instead? A structural and/or sequence alignment between these proteins might be nice.
- A good description of the metal binding site is lacking. Which residues coordinate the iron, their bonding distance and the metal coordination geometry is never discussed. The anomalous density at the Fe sites (Fig. S10) is extremely weak. It is comparable or even weaker than that of the Met sulfur even though the anomalous scattering from Fe at 1.59 Å wavelength should be ~ 6x greater. This indicates low occupancy. Was the Fe modeled at occupancy of 1? What was the B-factor? What does the 2Fo-Fc and Fo-Fc electron density look like at this site? Given the very high concentration of Fe used in co-crystallization, this looks more like adventitious binding than a high affinity binding site.
- The completeness for the Fe-IbpS at 86% is quite low. This table should be moved from SI to the main paper.

Minor Issues:

- Iron binding and dimerization residues are claimed to be "highly conserved" yet only IbpS, IbpP (which are nearly identical) and BcIbp are shown in the alignment in Fig. S9. Are these conserved across the whole Ibp family?
- No indication of statistical significance is given in Fig. 6C.
- Lines 338-339. "the atrbohD-atrbohF A. thaliana mutant, which is unable to produce H₂O₂ after a D. dadantii infection" should have a reference.
- Fig. 8A and C. The atrbohD-atrbohF mutant infected with WT D. dadantii appears to fare significantly better than the WT plant. Do the authors have any explanation for this? It should at least be acknowledged.
- Line 530. The Fe-IbpS needs a PDB accession code.
- Line 566. Are virulence scoring tests blinded? This should be indicated in the methods.

- Fig. 5. It would be helpful to show each monomer as a different color in order to see how close the metal binding site is to the dimer interface.
- There are a number of grammatical errors and typos. The manuscript would benefit from a careful proofreading.

Reviewer #2 (Remarks to the Author):

This paper presents evidence that the proteins IbpS and IbpP are secreted proteins capable of binding iron or copper, and that their production confers protection against oxidative stress and contributes to virulence. They also demonstrate that this family of metal binding proteins is present in the necrotrophic fungus *Botryotinia fuckeliana* (synonym *Botrytis cinerea*) and also contributes to virulence. Thus the novelty of this paper lies in its identification and characterisation of this new, broadly distributed group of metal binding proteins, with a role in host-microbe interactions.

The work presented in the paper is generally logical and reasonably well-described, but I have a few issues to raise. The evidence that IbpS provides protection against reactive oxygen is based on papers in which the purified protein was added to cultures of the wild-type bacteria. I was very surprised that the authors did not include data showing that the IbpS mutant bacteria showed reduced ROS tolerance *in vitro*, which would confirm that this protein conferred protection against ROS when produced at natural concentrations. If this experiment was performed and negative results were obtained the authors should include this information and some discussion of the results. A negative result would not necessarily overturn the authors' conclusions, it might indicate that the protein only accumulates to effective concentrations in diffusion-limited environments, which could be tested experimentally.

Instead of directly testing their hypothesis regarding the role of these proteins in oxidative stress tolerance the authors seek to corroborate their hypothesis by examining whether the difference in virulence between the mutant and wild-type bacteria is ablated in plants that do not show an oxidative burst. Although this evidence does support their hypothesis it does not definitively prove it, as these plants are altered in their ability to mount certain aspects of the plant immune response, which could provide alternate explanations for the results observed – e.g. the mutant bacteria show differential expression of genes needed to overcome rboh-mediated resistance – and in the absence of this resistance the differential phenotype disappears. This result should therefore be interpreted with appropriate caution. I also note that the mutant phenotypes that are described here are not validated by complementation, which should ideally be carried out to support the authors conclusions.

In addition qRT-PCR analyses are carried out using only a single housekeeping gene for normalisation, with no evidence of the validity of this gene as a suitable gene. Best practice would be to use 2-4 validated genes.

Specific comments

Ln 30-31 – phrasing can be improved.

Ln 32 “the” not “their”

Ln 77 “the role”

Ln 84 delete “by”

Ln 93 It would be useful to specify whether iron deficiency is observed in host, pathogen or both.

Ln 103 This statement slightly overstates the evidence presented, as the reduced virulence has not been directly linked to the impaired capacity of the organisms to detoxify ROS. This is inferred from various data.

Ln 133.Ln139 The exact conditions of the experiments in which chicory leaves are used to induce gene expression are not well-described. It would be useful to provide a precise indication of the

amount of chicory leaf tissue added per unit volume.

Ln 191 "but"

Ln 220/955 HEPES

Ln 257 specify figure

Ln 298 It is unclear why a concentration of 6 μM was used. Does this have biological relevance? The time point used for these experiments seems to be quite late to detect what could be a transient reduction in iron if IbpS is degraded by apoplastic proteases. Do the authors have any evidence to indicate how stable IbpS is in planta? Were other timepoints or concentrations tested?

Ln 303 delete "of"

Ln 317 The 30 mM concentration of hydrogen peroxide used for these experiments seems to be quite high. Is there a precedence for the use of this concentration in the literature? Was it optimised for this experiment?

Ln 354 "exhibit"

Ln 361 I am not convinced that the Ibp proteins should be termed effectors. This is a term that is being increasingly loosely and inappropriately applied within the plant pathology community. The term effector protein was originally applied to proteins secreted by the T3SS which interact with host targets to modulate host defences. This draws on the use of the term "effector molecule" for small molecules that selectively bind to proteins and regulate their activity. If the authors hypothesis for the function of IbpS is correct, then it does not conform to the sense of either of these definitions.

Ln 439 The description of the inoculum used is very vague. The authors should report the density of spores and the medium and mechanism by which they were applied to leaves.

Ln 450 "recombined in low phosphate medium" is very vague. This experiment should be described in more detail.

Ln 451 "checked by PCR" is vague. This should be described in more detail.

Ln 492 Specify the concentration at which amino acids were added

Ln 494 Provide more details on the mass spectrometry performed.

Ln 577 improve the phrasing of this line.

Ln 583/586 specify the concentration used.

Fig 1 A single phylogenetic tree is shown, thus the title should be "Phylogeny"

Fig 6. Specify the time/OD at which cultures were harvested.

Fig 6. 992 Does over 8 hours indicate at 8 h or the cumulative luminescence recording over 8 h?

Fig 8. It would be useful to add the results from statistical analysis of the data to panels a/b/c/d. Supplementary Fig 2 and Fig 3. As noted earlier, a more precise description of the chicory supplemented media is needed.

Supplementary Fig 2 "strain" not "train"

Reviewer #1 (Remarks to the Author):

Title: Reactive oxygen species protection of necrotrophic phytopathogens by a conserved secreted metal binding protein.

This manuscript describes the structure of a secreted protein (IbpS) related to ABC transporter solute binding proteins and its function in sequestering reactive metals to ameliorate oxidative stress in plant pathogens. This is an interesting and, to my knowledge, novel finding and seems to be applicable to a wide range of species. Gene expression and virulence assays are convincing and indicate that IbpS does protect the organism from oxidative stress and is important for full virulence. However, there are major problems with the in vitro metal binding assays, which suggest a very weak binding affinity for both Fe and Cu that cannot be physiologically relevant. Since the Ibp proteins are proposed to exert their antioxidant functions by metal binding, this is a major point that needs to be addressed. The analysis of structural data is also somewhat lacking. Specific points with regard to these issues are indicated below along with a few minor issues.

In vitro metal binding assays:

- The major problem with all of the assays is that saturation is never demonstrated. If a spectroscopic assay is to be used, a plot of fluorescence/absorbance versus [metal] must be shown and it must saturate. Whatever assay is used, binding affinity and stoichiometry for both metals should be determined. Also, why is there no data for IbpP?

New plots have been added (Supplementary Fig. 7). Fe titration was measured by UV-visible spectroscopy, no saturation was observed for up to 10 Fe equivalents. By measuring the fluorescence variation, and calculating $F_0 - F$, the resulting curve displays a biphasic behavior. Saturation occurred for 1 Fe equivalent, then a second slope was observed for increasing Fe concentration and saturation was never achieved. We deduced from these data that IbpS possesses a “high” affinity site and also encompasses non-specific binding. From the spectra it was not possible to calculate a K_D . However, given the concentration used, this K_D must be in the micromolar range. For that reason, we were unable to use metal competitors (such as EDTA) to remove Fe non-specifically bound, as the affinity of these compounds is much too high regarding the affinity of IbpS for Fe. We also attempted to perform ITC, however no satisfying results were

obtained, certainly in reason of non-specific binding. These observations are strengthened by the crystallographic data where weak binding of Fe to IbpS was observed when the experiments were conducted with large excess of Fe (see below). We apologize for the lack of K_D determination regarding Fe. However, we think that the in vitro experiments are convincing enough to demonstrate that IbpS binds Fe. These experiments are corroborated by the crystallographic data (new figures regarding Fe binding site were added in the revised version, and read below).

Moreover, we show in vivo that IbpS binds Fe, as intracellular concentrations of Fe decreased when IbpS was added to the medium (see Fig. 7)

Regarding IbpP, similar UV-visible and fluorescence spectra were recorded and not shown for the sake of manuscript length.

- ITC is probably the best way to quantify metal binding, but the data presented is of poor quality and only given for Cu. Here again, the titration is not run to saturation. Further, a blank run is not performed to indicate that the heat changes observed are not simply a result of dilution of the CuCl_2 solution.

A blank run was indeed performed in parallel, this point is now stated in text, in the Materials and Methods section. ITC experiments were performed again, changing the buffer in which IbpS was diluted (50 mM Tris-HCl, 100 mM NaCl pH (7) instead of 10 mM Hepes pH(7)). They now show saturation. The calculated K_D is in the sub-micromolar range. This is described in the text.

Crystallographic analysis:

- The authors state that IbpS does not contain the 4 Tyr and His residues that bind iron in FutA. What does it contain instead? A structural and/or sequence alignment between these proteins might be nice.

We apologise for our lack of clarity. In the previous manuscript we explained that the residues coordinating iron in IbpS structural homologs were absent because an entire helix is missing in our protein. We now provide a structural alignment of IbpS with a representative structure of Fe binding SBPs together with several representative IbpS homologs (Fig S8). We hope that this clarifies our text. We have also made a novel supporting figure (S9) to better show these structural differences.

- A good description of the metal binding site is lacking. Which residues coordinate the iron, their bonding distance and the metal coordination geometry is never discussed.

We agree that a description of the metal binding site is missing.

We now provide a detailed description of the metal binding site (see below).

The anomalous density at the Fe sites (Fig. S10) is extremely weak. It is comparable or even weaker than that of the Met sulfur even though the anomalous scattering from Fe at 1.59 Å wavelength should be ~ 6x greater. This indicates low occupancy. Was the Fe modeled at occupancy of 1? What was the B-factor? What does the 2Fo-Fc and Fo-Fc electron density look like at this site?

We apologise again for our lack of clarity. Thanks to her/his comment we have re-examined the structure and maps.

We do agree with reviewer 1 that the anomalous signal appears weaker than one could expect. However, we believe that this might look suspicious because of 1) poorly designed figure and 2) poor explanations.

The metal ions have indeed low occupancies Fe1 (oc 0.51, Bfac=23), Fe2 (oc 0.53, Bfac=25), Fe3 (oc 0.50, Bfac=23), Fe4 oc (0.49, Bfac=22) (this is now indicated in the text).

We have made better figures because the first one was misleading. The anomalous map indicates that the peak sigma for FE atoms is (on average) 9.5 and is 6.8 for the nearby Methionines 334 (see figure below of the 4 iron binding sites). So the anomalous sigma of iron is indeed weak but not weaker than the methionines.

The different maps (2Fo-Fc and Fo-Fc and anomalous) are now displayed in the new Fig.5.

Given the very high concentration of Fe used in co-crystallization, this looks more like adventitious binding than a high affinity-binding site.

We agree with referee 1 that it is not a high affinity site but we already stated that in the submitted manuscript (result section). This was confirmed by using the server check my metal (https://csgid.org/metal_sites). This server found that only a water molecule coordinates the iron and the metal is classified as poorly coordinated (as we said in the manuscript). By providing more details, as suggested by reviewer 1, we hope to have made this clearer in the revised version that we do not consider this site as a high affinity site.

We have revisited extensively our data and also performed additional experiments.

About the iron concentration used, we would like to point out our method section to the referee. After incubating the protein with high concentration of iron chloride (1.8 mM), the protein is 1) **washed** via desalting column (ZEBA) and then 2) crystallized **in absence** of Iron. Crystals were also washed again in cryo-solution **without** iron. So the concentration used is in fact not high but rather low for a low affinity binding protein.

Before collecting data on these crystals we had performed an energy scan on the Beamline at ESRF on this crystal (see below). The result clearly shows the presence of iron in the crystals with an absorption peak corresponding to Iron. It is noteworthy that the peak is not very high which also suggest partial occupation. We have used the same protocol (binding+crystallising+energy scan and crystal structure determination) to determine the structure of IbpS with Cu, Zn, or Mn. In all cases, we could not detect their presence in the crystal following the same procedure. We thus concluded that iron is indeed present, although with a low occupancy.

We have also tried to perform a different experiment using higher concentration of FeCl_3 (30 mM) and freeze the crystals in the presence of FeCl_3 (10mM). These crystals had a space group P21 with 4 molecules in the AU and diffract to 1.7Å resolution. The obtained structure showed anomalous density at the same positions as in Fe-IbpS (but not much stronger/higher) and also novel iron binding sites and large clusters of Fe bound to external residues of the protein

including E368, E212 and E209 (SEE figure below). These Fe correspond to large anomalous map peaks (>10 sigma) in the structure.

Crystal structure of IbpS in the presence of excess FeCl₃. In addition to the iron binding site observed in the Fe-IbpS structure, we can observe binding of iron clusters to external residues E209, E212 and E368. The anomalous map is contoured at 7 Sigma

These “external sites” disappeared if the crystals were back-soaked in a cryoprotectant not containing iron. We concluded that, these additional sites were likely adventitious, in contrast to the iron-binding site identified in Fe-IbpS structure. We would rather not include this structure in the present manuscript since 1) it does not provide additional information on the binding site and 2) these external sites are unlikely to be relevant.

In conclusion, we thank referee 1 for her/his questions that have helped us to provide stronger evidence of iron binding and better explanations. Although the iron-binding site is a low affinity site, it is still specific and we are confident that the metal in the structure is iron. We have now discussed this further in the revised version because it is possible that the *in vivo* ligand might be different or that structural arrangements are required to coordinate iron, for instance, via the triple aspartate motif.

- The completeness for the Fe-IbpS at 86% is quite low.

Indeed, unfortunately, this is due to the orientation of the crystal and given that it is a space 1 group, there is little we could do about it at this point.

This table should be moved from SI to the main paper.

We agree with the reviewer and have moved the table to the main manuscript.

Minor Issues:

- *Iron binding and dimerization residues are claimed to be “highly conserved” yet only IbpS, IbpP (which are nearly identical) and BcIbp are shown in the alignment in Fig. S9. Are these conserved across the whole Ibp family?*

We apologise for this and have now made a complete structural alignment with more sequences (Supplementary Fig. S8). We have also added a representative iron binding member of the cluster D family (FutA1) so that it is clearer to the reader that these residues are conserved across the family but not in FutA1 or homologs.

We have added a sentence explaining the method used to discriminate between strictly conserved (100%) and highly conserved (95%). We have used the same 122 sequences to generate the alignment (conservation) and the phylogenetic tree (Fig.1).

.

- No indication of statistical significance is given in Fig. 6C.

Results of statistical tests have been added (Note that this is now Fig.7c)

- Lines 338-339. “the atrbohD-atrbohF *A. thaliana* mutant, which is unable to produce H₂O₂ after a *D. dadantii* infection” should have a reference.

The reference has been added

- Fig. 8A and C. The atrbohD-atrbohF mutant infected with WT *D. dadantii* appears to fare significantly better than the WT plant. Do the authors have any explanation for this? It should at least be acknowledged.

Our results do not show that the mutant fares better than the WT since all the mutant *Arabidopsis* develop symptoms (Fig. 9c) while there are many WT *Arabidopsis* that develop no symptoms (Fig. 9a).

- Line 530. The Fe-IbpS needs a PDB accession code.

We have added the accession code for the Fe-IbpS structure.

- *Line 566. Are virulence scoring tests blinded? This should be indicated in the methods.*

Virulence scores are not blinded. This is now indicated in the methods.

- Fig. 5. It would be helpful to show each monomer as a different color in order to see how close the metal binding site is to the dimer interface.

We thank the reviewer for this suggestion. However, only a single chain is shown in the figure.

We have made a novel figure showing the two chains displayed in two colours. (Note that this is now Fig.6)

- There are a number of grammatical errors and typos. The manuscript would benefit from a careful proofreading.

We have corrected the text and hope it is more correct now.

Reviewer #2 (Remarks to the Author):

This paper presents evidence that the proteins IbpS and IbpP are secreted proteins capable of binding iron or copper, and that their production confers protection against oxidative stress and contributes to virulence. They also demonstrate that this family of metal binding proteins is present in the necrotrophic fungus *Botryotinia fuckeliana* (synonym *Botrytis cinerea*) and also contributes to virulence. Thus the novelty of this paper lies in its identification and characterisation of this new, broadly distributed group of metal binding proteins, with a role in host-microbe interactions.

The work presented in the paper is generally logical and reasonably well-described, but I have a few issues to raise. The evidence that IbpS provides protection against reactive oxygen is based on papers in which the purified protein was added to cultures of the wild-type bacteria. I was very surprised that the authors did not include data showing that the IbpS mutant bacteria showed reduced ROS tolerance in vitro, which would confirm that this protein conferred protection against ROS when produced at natural concentrations. If this experiment was performed and negative results were obtained the authors should include this information and some discussion of the results. A negative result would not necessarily overturn the authors' conclusions, it might indicate that the protein only accumulates to effective concentrations in diffusion-limited environments, which could be tested experimentally.

We thought to how we could realize this experiments but we could not find a satisfying way to realize it:

We induce IbpS synthesis by adding chicory pieces in the growth medium and after the overnight culture, the medium contains pieces of rotten tissues that could interact with H₂O₂. Moreover, to do the experiment we wash the bacteria, adjust the OD to 1 and dilute the bacteria 1000-fold. Dilution without washing of an induced culture of the WT or the mutant complemented strain will contain 1/1000 of the produced IbpS quantity. This is why we added exogenous IbpS to washed bacteria (to remove the rotten chicory tissue) and showed that it increases tolerance to H₂O₂. The experiment shown with *E. coli* confirms that the protection effect is due to IbpS addition.

We have estimated the concentration of IbpS in a culture in the presence of chicory pieces. It is about 0.5 μ M. We think that in the immediate proximity of the bacteria, the local concentration

could be much higher. This is why we used a 6 μ M concentration for the ROS protection experiment.

Instead of directly testing their hypothesis regarding the role of these proteins in oxidative stress tolerance the authors seek to corroborate their hypothesis by examining whether the difference in virulence between the mutant and wild-type bacteria is ablated in plants that do not show an oxidative burst. Although this evidence does support their hypothesis it does not definitively prove it, as these plants are altered in their ability to mount certain aspects of the plant immune response, which could provide alternate explanations for the results observed – e.g. the mutant bacteria show differential expression of genes needed to overcome rboh-mediated resistance – and in the absence of this resistance the differential phenotype disappears. This result should therefore be interpreted with appropriate caution. I also note that the mutant phenotypes that are described here are not validated by complementation, which should ideally be carried out to support the authors conclusions.

A comment has been added in the text stating that our results do not definitely prove a role of IbpS in ROS protection but that the effect observed could result from other defects in the Arabidopsis mutants.

Test of complementation of the *ibpS* mutant on Arabidopsis have been performed and are presented on Supplementary Fig. 11. Complementation could not be performed with *B. cinerea* but the characterization of a new mutant independently constructed shows the same reduced virulence as the one described in the text.

In addition qRT-PCR analyses are carried out using only a single housekeeping gene for normalisation, with no evidence of the validity of this gene as a suitable gene. Best practice would be to use 2-4 validated genes.

qRT-PCR analyses are now normalized with two genes (clathrin and PP2A) for Arabidopsis experiments and three genes (the actin-encoding gene (Bcin16g02020), the *bcefl* α gene (Bcin09g05760) and the *bcpda1* (Bcin07g01890) genes) for Botrytis experiments. This is indicated in the legend of the figures.

Specific comments

Ln 30-31 – phrasing can be improved.

Phrasing has been changed

Ln 32 “the” not “their”

Phrasing has been changed

Ln 77 “the role”

“The” has been added

Ln 84 delete “by”

“by” has been deleted

Ln 93 It would be useful to specify whether iron deficiency is observed in host, pathogen or both.

Iron deficiency response is induced in the plant. This is now indicated in the text.

Ln 103 This statement slightly overstates the evidence presented, as the reduced virulence has not been directly linked to the impaired capacity of the organisms to detoxify ROS. This is inferred from various data.

We have changed the sentence to “probably because of their impaired capacity to detoxify ROS produced during infection.”

Ln 133.Ln139 The exact conditions of the experiments in which chicory leaves are used to induce gene expression are not well-described. It would be useful to provide a precise indication of the amount of chicory leaf tissue added per unit volume.

We added about 2 g of chicory tissue in 5 ml of M63 medium. This has been added in Methods.

Ln 191 “but”

This part of the text has been changed

Ln 220/955 HEPES

Additional experiments have been performed in Tris-HCl buffer and HEPES is no longer used

Ln 257 specify figure

Fig 4c. Corrected

Ln 298 It is unclear why a concentration of 6 μM was used. Does this have biological relevance? The time point used for these experiments seems to be quite late to detect what could be a transient reduction in iron if IbpS is degraded by apoplastic proteases. Do the authors have any evidence to indicate how stable IbpS is in planta? Were other timepoints or concentrations tested? The concentration of 6 μM is similar to that used in the ROS resistance experiment. We have estimated the concentration of IbpS in a M63 liquid culture in the presence of chicory pieces to about 0.5 μM . We think that in the immediate proximity of the bacteria, the local concentration could be much higher. This is why we used a 6 μM concentration. Experiments were always performed 24h after inoculation.

When we analyzed the presence of IbpS in rotten plant tissues by Western Blot, no sign of protein degradation was observed. All the proteins secreted by the *D. dadantii* Type II secretion system are resistant to protease degradation.

Ln 303 delete “of”

Deleted

Ln 317 The 30 mM concentration of hydrogen peroxide used for these experiments seems to be quite high. Is there a precedence for the use of this concentration in the literature? Was it optimised for this experiment?

The concentration of 10 mM used for *D. dadantii* is that used by Reverchon et al. 2002 in the same type of experiment ([DOI: 10.1128/JB.184.3.654-665.2002](https://doi.org/10.1128/JB.184.3.654-665.2002)). The concentration of 30 mM for *E. coli* has been optimized for this experiment.

Ln 354 “exhibit”

Corrected

Ln 361 I am not convinced that the Ibp proteins should be termed effectors. This is a term that is

being increasingly loosely and inappropriately applied within the plant pathology community. The term effector protein was originally applied to proteins secreted by the T3SS which interact with host targets to modulate host defences. This draws on the use of the term “effector molecule” for small molecules that selectively bind to proteins and regulate their activity. If the authors hypothesis for the function of IbpS is correct, then it does not conform to the sense of either of these definitions.

The sentence has been changed to “suggest that most Ibp are secreted proteins”

Ln 439 The description of the inoculum used is very vague. The authors should report the density of spores and the medium and mechanism by which they were applied to leaves.

A new description of the inoculum has been added in methods

Ln 450 “recombined in low phosphate medium” is very vague. This experiment should be described in more detail.

Ln 451 “checked by PCR” is vague. This should be described in more detail.

Recombination and its verification are now described. A reference for the recombination technique has been added.

Ln 492 Specify the concentration at which amino acids were added

Concentration was added (0.01%)

Ln 494 Provide more details on the mass spectrometry performed.

Details on the protocol used to confirm the presence of three selenomethionines have been added.

Ln 577 improve the phrasing of this line.

The sentence has been changed.

Ln 583/586 specify the concentration used.

The concentrations are now specified.

Fig 1 A single phylogenetic tree is shown, thus the title should be “Phylogeny”

Corrected

Fig 6. Specify the time/OD at which cultures were harvested.

Now Fig 7. Overnight has been added

Fig 6. 992 Does over 8 hours indicate at 8 h or the cumulative luminescence recording over 8 h? The spectra were recorded during 8 hours in order to monitor the full kinetics of the expression of the gene fusion. What is displayed is the maximum specific activity (Luminescence / OD 600nm) occurring during the kinetics. In other words, we selected the peak of activity. By doing so, we get rid of possible offsets of gene expression between the different strains that not systematically grow at the same rate.

Fig 8. It would be useful to add the results from statistical analysis of the data to panels a/b/c/d.

Now Fig 9. Results of statistical analysis have been added on panel a/b and c/d

Supplementary Fig 2 and Fig 3. As noted earlier, a more precise description of the chicory supplemented media is needed.

A description of the chicory supplemented medium is given in Methods

Supplementary Fig 2 “strain” not “train”

Corrected

Reviewers' comments:

Reviewer #1 (Remarks to the Author):

Title: Reactive oxygen species protection of necrotrophic phytopathogens by a conserved secreted metal binding protein. Revision

The revision of this manuscript has satisfactorily addressed most of the issues raised in review, and the revised version is significantly improved in terms of presentation of crystallographic data. Binding of protein to Fe-NTA resin, taken together with clear iron incorporation into the crystal structure and depletion from the media is sufficiently convincing to indicate iron binding. That said, attempts at measuring iron binding by spectroscopic methods do not add much to the story, and the authors may consider simply removing these. Effective Fe(III) binding assays have been described for other iron solute binding proteins (e.g. Koropatkin et al. *J. Biol. Chem.* 282, 27468-77), and the authors may want to pursue these in future work. In addition, it may be worth mentioning that other ferric binding proteins bind a synergistic anion (Guo et al. *J. Biol. Chem.* 278, 2490-502), which may be required for high affinity binding. A couple of minor issues arose in the revised manuscript.

Lines 224-225: The following sentence has been added in revision: "However, the quality of the binding isotherm did not allow the determination of a K_D neither, certainly in reason of the non-specific binding of Fe." The word "neither" should be omitted or replaced with "either". Also, I do not understand what the second part of the sentence is trying to convey. Perhaps this should be replaced with "likely due to non-specific binding of Fe at high concentrations."

The axis labels on Fig. 2C are too small to be legible.

The version I have of Figures 3 and 4 have large white squares obscuring parts of the structures.

Line 563: The Fe-bound structure is reported as diffracting to 18 Å, rather than 1.8 Å,

Reviewer #2 (Remarks to the Author):

The authors have satisfactorily addressed the majority of my comments. However, I remain unconvinced by the authors' argument that it is not possible to directly assess whether a mutant strain lacking IbpS displays reduced ROS tolerance compared to a wild-type bacterium. They state that the addition of chicory is required to induce IbpS expression, and that the presence of chicory and its reaction with ROS could interfere with an in vitro experiment. However, if a standard amount of chicory tissue is added per culture, and bacterial numbers are determined by plating before and after the addition of hydrogen peroxide, then any reaction of the plant tissue with the hydrogen peroxide would occur equally in both the wild-type and mutant bacterial cultures, allowing a direct comparison of the effect of the mutation on ROS tolerance. If the authors believe it is necessary to separate the bacteria from the plant tissue, then they could mimic experiments by Deng et al. (2014) *Journal of Bacteriology* 196:2499-2513 in which the bacteria are contained within dialysis tubing, which could potentially allow IbpS expression to be induced by chemical signals emitted from the chicory and induced bacteria to be challenged with ROS.

It is notable that the authors state that in culture the concentration of IbpS is 0.5 μM, but in the paper their ROS tolerance experiments bacteria are treated with 50 μM IbpS (ln 1098), not 6 μM as stated in the rebuttal, and it remains unclear whether this is a biologically relevant concentration. The reason for performing the ROS tolerance assays in water rather than culture

media is also unclear. As this aspect of the experiment is specifically mentioned in the authors rebuttal this raises the question of whether the effect of ROS and IbpS treatments on bacterial viability could only be observed in water. Is water necessary to provide a sufficiently metal-ion limited environment such that an effect of IbpS can be observed? If so, then this also raises questions about whether the conditions used in this in vitro assay provide a valid proxy for the plant environment, where metal ions may be more abundant. The authors could consider repeating this experiment using apoplastic washing fluid to suspend the bacteria to provide a more biologically relevant environment.

In the absence of this data the statement in Ln 99/100 needs to be more qualified to be entirely accurate, and modified from "We demonstrated that by binding these metals, IbpS reduces the toxicity of H₂O₂, probably by preventing the Fenton reaction" to "We demonstrated that exogenously added IbpS reduces the toxicity of H₂O₂, probably by reducing the Fenton reaction"

Collectively, this paper presents interesting results that support the hypothesis that IbpS is a secreted protein that contributes to virulence in *D. dadantii*. However, its role in virulence remains speculative rather than being conclusively demonstrated by the results presented.

Minor corrections:

Ln 34 "limiting" or "restricting" could be a more accurate term than preventing, which suggest ROS accumulation is completely blocked.

Ln 124 "in an operon" or "in operons"

Ln 135 "the most"

Ln 154 "a hemibiotrophic"

Ln 157 "two other"

Ln 221 "and that"

Ln 346 "reduces" or "limits" rather than "prevents"

Reviewer #1 (Remarks to the Author):

The revision of this manuscript has satisfactorily addressed most of the issues raised in review, and the revised version is significantly improved in terms of presentation of crystallographic data. Binding of protein to Fe-NTA resin, taken together with clear iron incorporation into the crystal structure and depletion from the media is sufficiently convincing to indicate iron binding. That said, attempts at measuring iron binding by spectroscopic methods do not add much to the story, and the authors may consider simply removing these. Effective Fe(III) binding assays have been described for other iron solute binding proteins (e.g. Koropatkin et al. J. Biol. Chem. 282, 27468-77), and the authors may want to pursue these in future work. In addition, it may be worth mentioning that other ferric binding proteins bind a synergistic anion (Guo et al. J. Biol. Chem. 278, 2490-502), which may be required for high affinity binding. A couple of minor issues arose in the revised manuscript.

The part concerning spectroscopic methods has been deleted in the text (former lines 198 to 212), in Figure 2 (Fig 2b and 2c) and in supplementary Fig 7 (Suppl Fig 7a). Suppl Fig 7d has been moved to Fig 2b.

A sentence on the possibility of a synergistic anion has been added l 286

Lines 224-225: The following sentence has been added in revision: “However, the quality of the binding isotherm did not allow the determination of a KD neither, certainly in reason of the non-specific binding of Fe.” The word “neither” should be omitted or replaced with “either”. Also, I do not understand what the second part of the sentence is trying to convey. Perhaps this should be replaced with “likely due to non-specific binding of Fe at high concentrations.”

The end of the sentence has been changed to « likely due to non-specific binding of Fe at high concentrations » (now l 210)

The axis labels on Fig. 2C are too small to be legible.

Fig. 2C has been deleted

The version I have of Figures 3 and 4 have large white squares obscuring parts of the structures. New pdf of Figures 3 and 4 have been uploaded with the manuscript

Line 563: The Fe-bound structure is reported as diffracting to 18 Å, rather than 1.8 Å,
Now line 558. This has been corrected.

Reviewer #2 (Remarks to the Author):

The authors have satisfactorily addressed the majority of my comments. However, I remain unconvinced by the authors' argument that it is not possible to directly assess whether a mutant strain lacking IbpS displays reduced ROS tolerance compared to a wild-type bacterium. They state that the addition of chicory is required to induce IbpS expression, and that the presence of chicory and its reaction with ROS could interfere with an in vitro experiment. However, if a standard amount of chicory tissue is added per culture, and bacterial numbers are determined by plating before and after the addition of hydrogen peroxide, then any reaction of the plant tissue with the hydrogen peroxide would occur equally in both the wild-type and mutant bacterial cultures, allowing a direct comparison of the effect of the mutation on ROS tolerance. If the authors believe it is necessary to separate the bacteria from the plant tissue, then they could mimic experiments by Deng et al. (2014) *Journal of Bacteriology* 196:2499-2513 in which the bacteria are contained within dialysis tubing, which could potentially allow IbpS expression to be induced by chemical signals emitted from the chicory and induced bacteria to be challenged with ROS.

It is notable that the authors state that in culture the concentration of IbpS is 0.5 μM, but in the paper their ROS tolerance experiments bacteria are treated with 50 μM IbpS (ln 1098), not 6 μM as stated in the rebuttal, and it remains unclear whether this is a biologically relevant concentration. The reason for performing the ROS tolerance assays in water rather than culture media is also unclear. As this aspect of the experiment is specifically mentioned in the authors rebuttal this raises the question of whether the effect of ROS and IbpS treatments on bacterial viability could only be observed in water. Is water necessary to provide a sufficiently metal-ion limited environment such that an effect of IbpS can be observed? If so, then this also raises questions about whether the conditions used in this in vitro assay provide a valid proxy for the

plant environment, where metal ions may be more abundant. The authors could consider repeating this experiment using apoplastic washing fluid to suspend the bacteria to provide a more biologically relevant environment.

- We performed the experiment exactly as suggested by Reviewer 2. Bacteria were grown in Tris minimal medium in the presence of chicory pieces to induce IbpS synthesis. In this condition, an important difference in the survival rate to H₂O₂ stress was observed showing the role of IbpS (Fig. 8c). This effect is reversed in a strain containing a plasmid bearing *ibpS* confirming the role of IbpS (for the level of production of IbpS in the complemented strain, see Supplementary Fig. 10). Moreover, this experiment was performed in Tris minimal medium + chicory which shows that the effect is not only visible in water.

The ROS resistance has also been performed in water because it is a protocol usually used to test the effect of siderophores on ROS resistance (Dellagi A et al. *Mol Plant Microbe Interact* **11**, 734-742 (1998)). We have performed new experiments using lower, more physiological, concentrations of IbpS (0.5 and 5 μM). They show that at these concentrations, IbpS has a H₂O₂ stress-protection effect (see Fig. 8 a and b).

We also tried to grow bacteria in a dialysis tubing. No induction of IbpS was observed, probably because induction is contact-dependent or because the inducer cannot be dialyzed.

Minor corrections:

Ln 34 “limiting” or “restricting” could be a more accurate term than preventing, which suggest ROS accumulation is completely blocked.

Preventing has been changed to limiting

Ln 124 “in an operon” or “in operons”

Corrected

Ln 135 “the most”

Corrected

Ln 154 “a hemibiotrophic”

Corrected

Ln 157 “two other”

Corrected

Ln 221 “and that”

text deleted

Ln 333 (ex 346) “reduces” or “limits” rather than “prevents”

Prevents has been changed to limits

REVIEWERS' COMMENTS:

Reviewer #2 (Remarks to the Author):

I am satisfied that the amendments made to this manuscript have addressed my comments, and have no further comments at this time.